# β subunits of GABA$_A$ receptors form proton-gated chloride channels: Insights into the molecular basis

Aleksandra Garifulina [1✉], Theres Friesacher [1], Marco Stadler[1], Eva-Maria Zangerl-Plessl [1], Margot Ernst [2], Anna Stary-Weinzinger [1], Anita Willam[1,3] & Steffen Hering [1,3✉]

Gamma-aminobutyric acid type A receptors (GABA$_A$Rs) are ligand gated channels mediating inhibition in the central nervous system. Here, we identify a so far undescribed function of β-subunit homomers as proton-gated anion channels. Mutation of a single H267A in β3 subunits completely abolishes channel activation by protons. In molecular dynamic simulations of the β3 crystal structure protonation of H267 increased the formation of hydrogen bonds between H267 and E270 of the adjacent subunit leading to a pore stabilising ring formation and accumulation of Cl$^-$ within the transmembrane pore. Conversion of these residues in proton insensitive ρ1 subunits transfers proton-dependent gating, thus highlighting the role of this interaction in proton sensitivity. Activation of chloride and bicarbonate currents at physiological pH changes (pH$_{50}$ is in the range 6- 6.3) and kinetic studies suggest a physiological role in neuronal and non-neuronal tissues that express beta subunits, and thus as potential novel drug target.

[1] Division of Pharmacology and Toxicology, Department of Pharmaceutical Sciences, University of Vienna, A-1090 Vienna, Austria. [2] Department of Pathobiology of the Nervous System, Medical University of Vienna, A-1090 Vienna, Austria. [3] ChanPharm GmbH, Am Kanal 27, Top 2/3/5, 1110 Vienna, Austria. ✉email: aleksandra.garifulina@univie.ac.at; steffen.hering@univie.ac.at

γ-Aminobutyric acid (GABA) type A receptors (GABA$_A$R) are ligand gated chloride channels mediating fast synaptic inhibition in the central nervous system (CNS)[1] and are expressed in most tissues[2]. GABA$_A$R are pentamers that potentially can be assembled of 19 possible subunits (α(1–6), β(1–3), γ(1–3), δ, ε, θ, π and ρ(1–3))[3,4] in mammalians. The most abundant native receptors are heteropentamers consisting of two α1, two β2 and one γ2 subunit[5]. A large number of receptor subtypes comprising different subunit combinations is, however, likely to exist[5]. In heterologous expression systems β3 and ρ1 subunits form functional homomers[6–8].

The crystal structure of a homopentameric β3 receptor provided insights into the molecular architecture of pentameric ligand gated ion channels[9]. This study, together with the crystal structure of a GLIC-GABA$_A$R α1 subunit chimera[10], and cryoelectron microscopy structures revealed detailed pictures of the extracellular domain and associated GABA binding loops at the β + /α− subunit interface, the central anion-conducting pore region in the TMD, potential locations of the channels activation and desensitisation gates and locations of drug binding sites[9,11–14].

Homomeric β3 receptors are lacking the β + /α− subunit interface and are therefore not sensitive to GABA. However, these receptors share properties with heteropentamers such as picrotoxin-sensitivity and contain binding determinants for histamine, propofol, pentobarbital and other ligands, which makes them interesting research subjects[7,15–19].

The receptors subunit composition affects the kinetics of chloride currents (I$_{GABA}$), the potency of their agonists and the pharmacological properties[20]. GABA$_A$Rs are targets for many drugs (such as benzodiazepines, barbiturates, propofol and volatile anaesthetics) that change I$_{GABA}$ in a subunit-specific manner[21–23].

Subunit-specificity is also a hallmark of GABA$_A$R modulation by protons. Acidification of the external solution differentially potentiates I$_{GABA}$ through αβ-, αβδ- and αβγδ- receptors[24,25]. A histidine residue (H267) is essential for proton-induced increase of I$_{GABA}$ through α1β1 or α1β2-receptors[26]. Hence, I$_{GABA}$ potentiation is abolished if this residue on the β2 subunit in the TM is substituted (H267A[26]). Site-specific mutagenesis later revealed that a lysine residue, K279 in the β subunit TM2–TM3 linker, is important for I$_{GABA}$ modulation by alkaline pH[25].

Here we made the surprising observation that homomeric receptors composed of either mutant β1(S265N), β2 or β3 subunits form chloride channels that are directly activated by protons. Proton-activated chloride currents (I$_{H(β)}$) through homomeric receptors formed by GABA$_A$ β subunits activate and desensitise with slower kinetics than currents through GABA$_A$ receptors (I$_{GABA}$).

In view of the essential role of the protonatable histidine (H267) on β-subunits and the available structural information[9] we probed its role and potential interactions with adjacent residues on pore forming TM2 in β-subunit homomers. The mutation H267A completely abolished proton sensitivity. Molecular dynamic simulations and functional studies revealed a key role of H267 and E270 interactions. Furthermore, proton sensitivity of β homomers can be transferred if these two conserved residues of β-subunits are inserted into homologous positions of the insensitive ρ1. The proton sensitivity of the resulting gain-of-function double mutant ρ1(G331H-A334E) was comparable to the one of β homomers. Jurkat cells feature a pictrotoxin sensitive, amiloride insensitive component of proton-induced currents. Overall, our study reveals a novel function of the GABA$_A$ β-subunits, the formation of proton-gated ion channels.

Mutational studies and MD simulations of a crystal structure disclosed the essential role of histidine-glutamate interaction in proton-sensing of this novel channel type.

## Results

**Homomers of GABA$_A$ receptor β subunits form proton-gated chloride channels.** Application of protons in concentration jumps from pH 9 to pH valued between 7 and 4 to oocytes expressing either human or rat GABA$_A$ β3 subunits induced

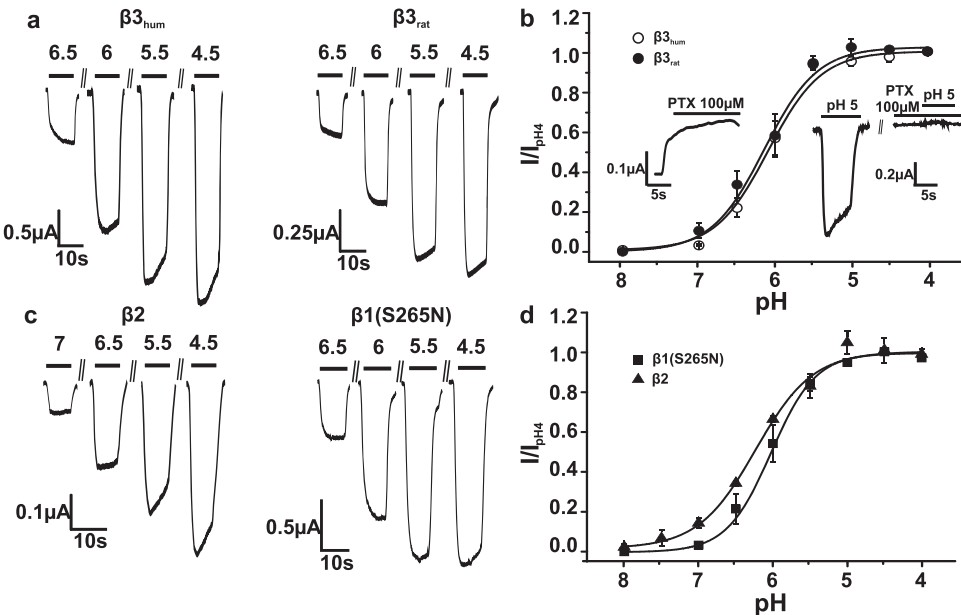

**Fig. 1 Proton-induced currents (I$_{H(β)}$) through homomeric GABA$_A$ β1(S265N), β2 and β3 receptors induced at a holding potential of -70 mV from pH of 9. a, c** Representative I$_{H(β)}$ through β3 (human, rat); β1 and β2 subunit homomers elicited by rapidly switching the pH from 9 to the indicated values. **b, d** Normalized pH-response curve of I$_{H(β)}$ through β3$_{hum}$ (pH$_{50}$ = 6.09 ± 0.2, $n_H$ = 1.25 ± 0.37), β3$_{rat}$ (pH$_{50}$ = 6.1 ± 0.01, $n_H$ = 1.25 ± 0.08), β2 (pH$_{50}$ = 6.23 ± 0.13, $n_H$ = 1.07 ± 0.05) and β1(S265N) (pH$_{50}$ = 6.03 ± 0.1, $n_H$ = 1.45 ± 0.21) homomers. The data are presented as mean values ±SEM, n = 5. The insets in (**b**) illustrate block of baseline current (left) and inhibition of I$_{H(β)}$ (right).

inward currents ($I_{H(\beta)}$) (Fig. 1a) whose amplitudes increased with increasing proton concentration.

Expression of GABA$_A$ β3 subunits in *Xenopus* oocytes resulted at pH = 7.4 in significant base line currents that were blocked by picrotoxin (Fig. 1b, inset). Raising the pH of the external solution to pH 9 reduced these currents significantly and pH 9 was thus used for most recordings of base line currents.

$I_{H(\beta 3)}$ activated with half-maximal activation (pH$_{50}$) of 6.09 ± 0.2 (β3 human) and 6.1 ± 0.01 of (β3 rat) and saturation at pH 5 (see Fig. 1a for typical current recordings and Fig. 1b for pH-response curves). Like baseline currents, proton-activated currents were inhibited by picrotoxin (Fig. 1b, inset). The current reversal potential indicates that chloride ions are the main charge carrier (Supplementary Fig. 1).

Proton-gated ion channel were also formed in mammalian cells. Making use of planar whole-cell patch clamp of CHO cells expressing human β3 GABA$_A$R we observed pH-induced currents activating with a pH$_{50}$ of 6.03 ± 0.19 (Supplementary Fig. 2).

Subsequent experiments revealed that other β subunits also form proton-gated channels. Figure 1c (left) illustrates similar activation of $I_{H(\beta 2)}$ at pH < 7 with pH$_{50}$ 6.23 ± 0.13 (β2). However, large picrotoxin-sensitive baseline currents of up to several µA (at pH 7.4) in oocytes expressing wild type β1 subunits first prevented activation of $I_{H(\beta 1)}$. These currents were reduced by shifting the pH to 9 but even under these conditions were too large for inducing $I_{H(\beta 1)}$. We therefore made use of mutation S265N that is known to reduce constitutive activity of β1 homomers[27]. $I_{H(\beta 1(S265N))}$ activated at pH < 7.5 with pH$_{50}$ of 6.03 ± 0.1 and served further as tool mutation to study $I_{H(\beta 1)}$ (Fig. 1c, d).

**Functional properties of proton-activated chloride channels.** Proton induced chloride currents through β3 homomers activate and desensitise with a slower time course than $I_{GABA}$ through heteropentameric α1βx receptors. This is exemplified in Fig. 2a, b comparing current kinetics of $I_{H(\beta 3)}$ and $I_{GABA}$ (α1β3 subunit heteropentamers) at saturating agonist concentrations (pH 5 and 100 µM GABA (pH$_{GABA}$ 7.2)). Mean time to peak were 1.52 ± 0.06 s ($I_{H(\beta 3)}$) and 0.63 ± 0.08 s ($I_{GABA}$) ($p < 0.05$, $n = 5$) and corresponding mean desensitisation half-times amounted 48.65 ± 5.92 s and 32.16 ± 2.77 ($p < 0.05$, $n = 5$), respectively.

The steady-state desensitization of β homomers occurs with comparable mid points (pH of half-maximal desensitisation for β1(S265N) at 6.4 ± 0.02, β2 at 6.97 ± 0.06 and β3 at 6.6 ± 0.15, Fig. 2d). Superimposed activation and desensitisation curves illustrate that at a physiological pH a significant fraction of these channels is expected to sojourn in an activated non-desensitised state. This finding is in line with the observed picrotoxin sensitive "baseline" or "leak" currents at pH 7.4 in oocytes expressing β homomers (Fig. 1b, inset). A decrease in extracellular pH from 7.2 to 6.5 induces significant $I_{H(\beta)}$ (exemplified for β3 homomers Fig. 2c).

Proton-activated currents were, however, also observed in oocytes expressing heteropentameric receptors comprising either α1β1, α1β2 or α1β3 subunits (Supplementary Fig. 3). pH-dependent activation of these currents occurred with similar kinetics and pH dependence as $I_{H(\beta x)}$ (Supplementary Fig. 3) suggesting current flow through β homomers that are formed in addition to the heteropentamers. $I_{H(\alpha 1\beta x)}$ activated with pH$_{50}$ values of pH$_{50}$ = 5.76 ± 0.04, $n_H$ = 1.36 ± 0.11 (α1β1), pH$_{50}$ = 6.14 ± 0.09, $n_H$ = 1.09 ± 0.14 (α1β2) and pH$_{50}$ = 6.06 ± 0.03, $n_H$ = 1.73 ± 0.14 (α1β3).

Subunit concatenation (α1-β2/β2-α1-β2, α1-β3/β3-α1-β3 and α1-β3-α1/γ2-β3) prevented both the formation of β homomers and at the same time activation of $I_{H(\beta)}$ (Supplementary Fig. 4).

**Pivotal role of H267 in proton activation of β3 homomers.** In heterologous GABA$_A$ receptors allosteric modulation of $I_{GABA}$ by protons is largely attributed to protonation of a single histidine (H267 Fig. 3a[26]). In view of the strong effect of mutation H267A that completely prevents allosteric pH modulation of $I_{GABA}$[26] it was tempting to analyse the impact of this residue on proton activation of β homomers.

As illustrated in Fig. 3b mutation H267A completely abolished activation of $I_{H(\beta)}$ in β3 human homomers. Instead, switching the pH from 9.0 to 5 induced a reversible and pH-dependent inhibition of the baseline current. These data indicate that protonation of histidine in position 267 is not only essential for allosteric pH modulation of heteromeric GABA$_A$ receptors but simultaneously has a key role in direct activation of β homomers by protons.

Structural analysis of the human homomeric GABA$_A$ β3 receptor[9] identified the interaction of H267 with a glutamate in position 270 at the neighbouring subunit. The motif HxxE is specific to β subunits and absent in other GABA$_A$ receptor subunits (Fig. 3c). The authors conclude that a ring of salt-bridges involving H267 and E270 from adjacent M2 helices contributes to stabilization of the extracellular portion of the pore in GABA$_A$R-β3cryst (see[9] for details).

To further investigate this interaction, we introduced several point mutations into position E270 of β3 subunits. However, mutation of E270 to alanine and glutamine resulted in non-functional channels. Furthermore, exchanging H267 and E270 (resulting in double mutant H267E-E270H) or other substitutions of H267 (H267K, H267R) did not lead to formation of functional homomers.

To further investigate the role of H267 (Fig. 3a) in proton activation, we performed 2 µs long MD simulations of the GABA$_A$ receptor, carrying a protonated (GABA$_A$-prot) and deprotonated (GABA$_A$-deprot) form of H267, respectively. Protonation caused H267 to contact the residue E270 of the neighbouring subunit via hydrogen bonds (Fig. 3d). For GABA$_A$-prot, a substantial fraction of the simulation frames displayed 5 and more hydrogens bonds between these residues (Fig. 3d), which induced the formation of a ring-like structure at the apex of the transmembrane domain (TMD) of the pore (Fig. 3e) (see also[9]). This interaction was drastically reduced for GABA$_A$-deprot, which features 0–2 hydrogen bonds in most of the simulation frames and hence lacks the ring pattern (Fig. 3d, e).

**Transfer of proton sensitivity to a ρ1 subunit.** Further evidence for a crucial role of H267 - E270 interaction in proton activation was obtained when the corresponding residues of β3 subunits were inserted into a ρ1 subunit. Homomeric receptors formed of ρ1 subunits are not activated by protons (Fig. 4a). Conversion of both residues (G331H-A334E) resulted, however, in transfer of proton sensitivity with a comparable pH$_{50}$ of 5.66 ± 0.1 (ρ1(G331H-A334E)) (Fig. 4c, d) while the sole transfer of the histidine (G331H) was insufficient (Fig. 4b).

**Conformational changes induced by protonation of H267.** In order to investigate further effects of H267 protonation on channel behaviour, we analysed the Cl$^-$ density, the pore radius and the hydrophobicity of the pore-facing residues in the two simulations (Fig. 5). The most striking difference between the two systems can be seen for the Cl$^-$ density. While GABA$_A$-prot is well-populated with Cl$^-$ nearly throughout the entire pore, we see an almost complete depletion of the Cl$^-$ density in the TMD of GABA$_A$-deprot. Another difference can be seen in the hydrophobicity of the pore-facing residues. Especially the region around and below the desensitization gate is more hydrophobic

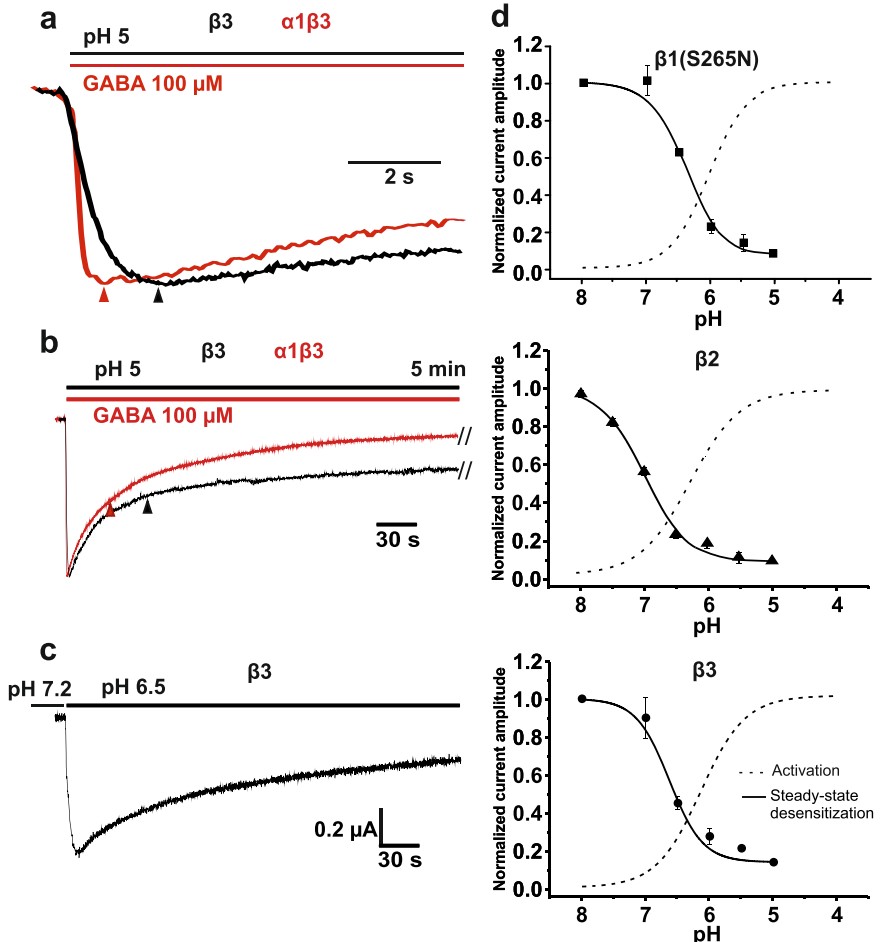

**Fig. 2 Functional properties of proton-activated homomeric β receptors. a** Activation kinetics of $I_{H(\beta3)}$ (pH 5) through human homomeric β3 receptors and $I_{GABA}$ (GABA 100 μM, pH 7.2) through α1β3 GABA$_A$ receptors at saturating agonist concentrations. The current traces illustrate slower activation of $I_{H(\beta3)}$ ($t_{peak} = 1.91$ s) compared to $I_{GABA}$ ($t_{peak} = 0.68$ s). **b** Desensitisation kinetics of $I_{H(\beta3)}$ and $I_{GABA}$ during long lasting exposure to pH 5 or to GABA 100 μM. Desensitization half time for $I_{H(\beta3)}$ is $t_{0.5} = 58.5$ s compared to $t_{0.5} = 33.6$ s of $I_{GABA}$. pH in (**a, b**) was changed from 9 to 5. **c** Representative $I_{H(\beta3)}$ induced by switching the pH from 7.2 to 6.5. **d** The pH-dependence of steady-state desensitization and of activation of receptors composed of the indicated β-subunits. Oocytes were conditioned for 10 min at the indicated pH, and receptors were subsequently activated by shifting the pH rapidly to 4.5 in order to estimate the the fraction of available (not yet desensitized) channels. The normalized current response is plotted as a function of the conditioning pH (superimposed data for current activation (dash lines) are taken from Fig. 1). Half-maximal desensitisation occurs at pH 6.4 ± 0.02 for β1(S265N), at pH 6.97 ± 0.06 for β2 and at pH 6.6 ± 0.15 ($n = 4$) for β3 homomers. pH was reduced from 9 to the indicated values. The data are presented as mean values ±SEM, $n = 5$.

in case H267 is deprotonated. Furthermore, the protonation state of H267 affects the pore diameter of the TMD. For GABA$_A$-prot we see a narrowing of the channel, stretching from the apex of the TMD, where H267 is located, to the desensitization gate at the lower end of the pore. The difference in pore radius between the two systems reaches up to ~2 Å, observed below H267 and at the desensitization gate.

**Picrotoxin sensitive, proton-induced currents in Jurkat cells.** Two cell systems were studied to gain first insights into a potential physiological role of $I_{H(\beta X)}$. In Jurkat cells (an immortalised line of human T lymphocyte cells) that are known to express predominantly β-subunits of GABA$_A$ receptors[28,29], we observed a fast activating (rapidly desensitising) inward current when pH was shifted from 7.2 to 5.0 (Supplementary Fig. 5a). This current was partially blocked by amiloride (200 μM) suggesting current through ASIC[30]. A remaining current in the presence of amiloride activated with slower kinetics and was completely blocked by 200 μM picrotoxin suggesting $I_{H(\beta x)}$ through proton-sensitive β-homopentamers in these cells. GABA

(100 μM) did not activate chloride currents (Supplementary Fig. 5a, right panel) which is in line with the higher RNA levels of β subunits compared to those of α subunits[28,29].

In iPS cell-derived GABANeurons shifting the pH to 5.0 also induced a rapidly activating ASIC-like current that was, however, completely inhibited by amiloride (200 μM, Supplementary Fig. 5b). Activation of $I_{GABA}$ in these cells indicated expression of heteromeric GABA$_A$ receptors which is known to reduce the formation of homooligomeric receptors[31,32] and would explain the absence of residual picrotoxin-sensitive currents.

**Discussion**

In this study we identified a previously unknown function of GABA$_A$ receptor beta subunits – the formation of proton-activated homomeric anion channels (Fig. 1, Supplementary Fig. 2). Considering the many roles of pH in physiological processes, proton-gated anion channels can readily be envisioned to play many roles in rapid responses to changes in pH, or even to contribute to pH stabilization by way of their bicarbonate conductance.

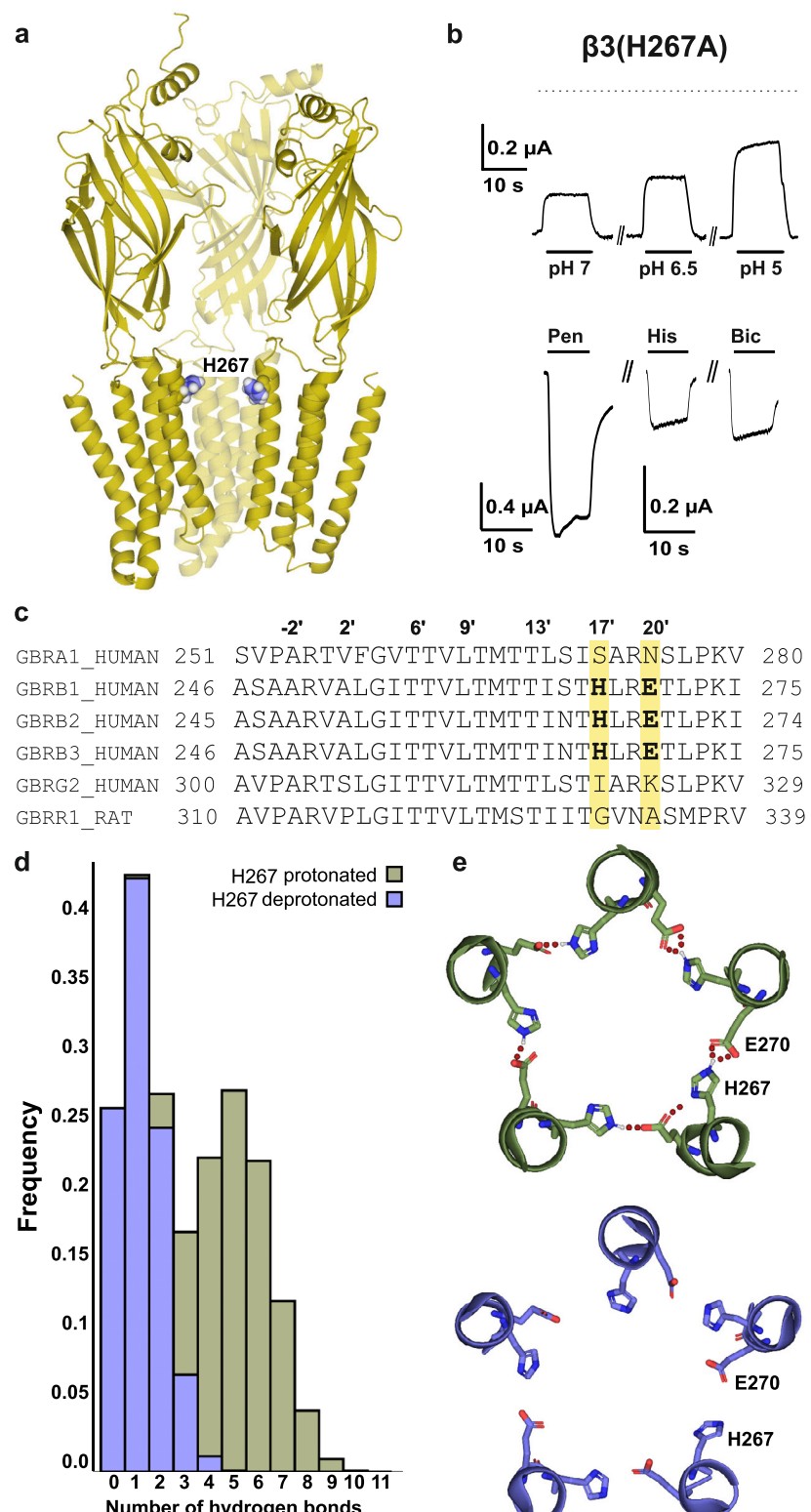

I$_{H(\beta x)}$ of all three receptor subtypes activate in *Xenopus oocytes* at pH < 7 with pH$_{50}$s ranging from 6.03 ± 0.1 (β1(S265N)) to 6.23 ± 0.02 (β2) (Fig. 1a–d). Proton-gated anion channels with similar pH sensitivity were also formed in mammalian cells expressing β3 subunits (Supplementary Fig. 2). It is currently not clear if the observed left shift of I$_{H(\beta1(S265N))}$

activation curve and its steeper slope result from mutation S265N or represent higher proton sensitivity of β1 homomers (Fig. 1d). Acidifications at physiological pH (e.g. pH changes from 7.2 to 6.5) activate I$_{H(\beta x)}$ which highlights the high proton sensitivity of these channels (illustrated for I$_{H(\beta3)}$ in Fig. 2c, see also Supplementary Fig. 2).

**Fig. 3 Loss of function (β3(H267A)) mutation in GABA$_A$ subunit homomers and a hydrogen bond formation between H267 and E270. a** Location of H267 in the pore region of the crystal structure of β3 homomers (the position of H267 is indicated with violet spheres)[9]. **b** Mutation H267A in the human subunits β3 subunit prevented activation of I$_{H(β3(H267A))}$ but, instead, induced pH-dependent inhibition of large baseline currents (upper traces). Dotted line illustrates zero current level. Lower traces illustrate retained sensitivity for other agonists (100 μM pentobarbital, 1 mM histamine and 100 μM bicuculline). **c** Sequence alignment of the transmembrane M2 α-helix of α1, β1–3 (human), γ2 (human) and ρ1 (rat) subunits (segment does not differ between rat and human). The location of H267 and E270 in β subunits and the corresponding homologous positions in the ρ1 subunit (double mutant ρ1(G331H-A334E)) are illustrated. **d** Histogram of the number of hydrogen bonds between protonated and deprotonated H267 and E270. **e** Snapshots of the ring structure observed in the MD runs.

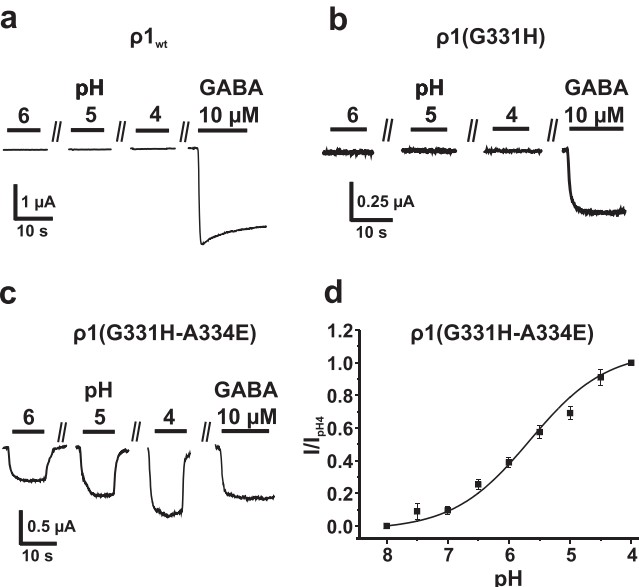

**Fig. 4 Gain of function mutations in GABA$_A$ ρ1(G331H-A334E) subunit homomers. a** Homomers of wild type ρ1 subunits are insensitive to protons but activated by GABA (10 μM, pH 7.2). **b, c** A single point mutation ρ1(G331H) was ineffective while double mutation ρ1(G331H-A334E) transferred proton sensitivity. **d** Normalized peak current values of I$_{H(ρ1G331H-A334E)}$ at indicated pH fitted to the Hill equation (pH$_{50}$ = 5.66 ± 0.1, $n_H$ = 0.7 ± 0.12). The data are presented as mean values ±SEM, $n$ = 9. pH changes in all experiments from 9 to indicated values.

I$_{H(βx)}$ activate slower than I$_{GABA}$ with kinetics and proton sensitivities resembling the properties of bacterial GLIC and other member of pentameric bacterial ligand gated chloride channels (Fig. 2a–c, also see[33]).

Proton-induced activation and steady-state desensitisation revealed a window I$_{H(βx)}$ in the pH range between 7 and 6 (Fig. 2d). This current corresponds to a picrotoxin-sensitive leak chloride conductance that was first observed in oocytes expressing β1 homomers[34] and later described for other β subunits[7,24].

β homomers are directly activated by protons and are thus clearly distinguishable from the known positive allosteric modulation of I$_{GABA}$ observed upon co-application of GABA and protons[26]. Proton-induced currents in oocytes expressing αβ heteropentamers display similar activation curves and pH$_{50}$ as observed for I$_{H(βx)}$ and are completely prevented by receptor concatenation (Supplementary Fig. 3, Supplementary Fig. 4). Both observations support the hypothesis that proton-activated currents in oocytes expressing heteropentamers result from the formation of a population of homopentamers[35].

Intrigued by the key role of H267 (Fig. 3a) in allosteric potentiation of GABA$_A$ receptors and structural data suggesting the formation of salt bridges between H267 and E270 at β3 subunit interfaces we substituted H267 in β homomers by

alanine and investigated proton activation. Interestingly, H267A not only eliminated proton-induced receptor activation but additionally induced large picrotoxin-sensitive baseline currents that were blocked by protons, indicating the existence of molecular mechanisms for proton activation and inhibition in β homomers (Fig. 3b).

Substitutions of E270 by alanine or glutamine, exchanging H267 and E270 (mutant H267E-E270H) or other substitutions of H267 (H267K, H267R) did not lead to formation of functional channels, which prevented more detailed analyses of this interaction.

However, transfer of both residues to a proton insensitive ρ1 subunit resulted in gain of function, indicating a potential key role of the interaction of these residues in proton-induced gating (Fig. 4a, b).

In order to elucidate the effect of H267 protonation in proton-dependent gating, we conducted MD simulations of the human GABA$_A$ β3 homopentamer (PDB: 4COF) exhibiting different protonation states at position 267 (Fig. 3a). Our simulations with GABA$_A$-prot unravel a ring-like structure at the apex of the transmembrane domain (TMD), which is stabilized by a formation of hydrogen bonds between protonated H267 and E270 of adjacent subunits (Fig. 3d, e). This interaction is accompanied by a narrow transmembrane pore with especially small radii at the potential desensitization gate[9,36]. The conformations adopted by GABA$_A$ β3$_{prot}$ in our simulations are therefore reminiscent of the initial crystal structure 4COF[9], which represents the agonist-bound, desensitized state of the receptor. The dual-gate model by Gielen and Corringer[36] proposes that desensitization of pLGICs is facilitated by a closing of the cytoplasmic end of the TMD. According to this model, our simulations with GABA$_A$ β3$_{prot}$ are likely to represent a post-activated, desensitized state. Importantly, comparison of inhibitory Cys-loop receptors[37] in putative shut and open conformations identified a dilation at the cytoplasmic end of the pore to be associated with a closed channel conformation. Similarly, deprotonation of H267 causes a widening of the TMD and a relief of the constriction at the desensitization gate in our simulations.

Our computational investigations suggest an involvement of H267 protonation in switching between different gating conformations, and thus corroborate our experimental findings of an activation of GABA$_A$ receptors by protonation.

Our simulations with GABA$_A$ β3$_{deprot}$ displayed an exclusion of Cl$^-$ from the TMD, while GABA$_A$ β3$_{prot}$ showed sizeable Cl$^-$ densities in both the extracellular and transmembrane domain. This finding further supports our experimental results of a receptor activation by protons and is reminiscent of a computational study on the glycine receptor (GlyR)[38], which reported Cl$^-$ permeation though the transmembrane pore in the open structure, while the same region was cleared from Cl$^-$ in the closed conformation. Small differences in the hydrophobicity of the pore lining residues might be partially accountable for the different ion occupancies. However, more sampling, which go beyond the scope of this study, might be needed to make a hydrophobic gate visible.

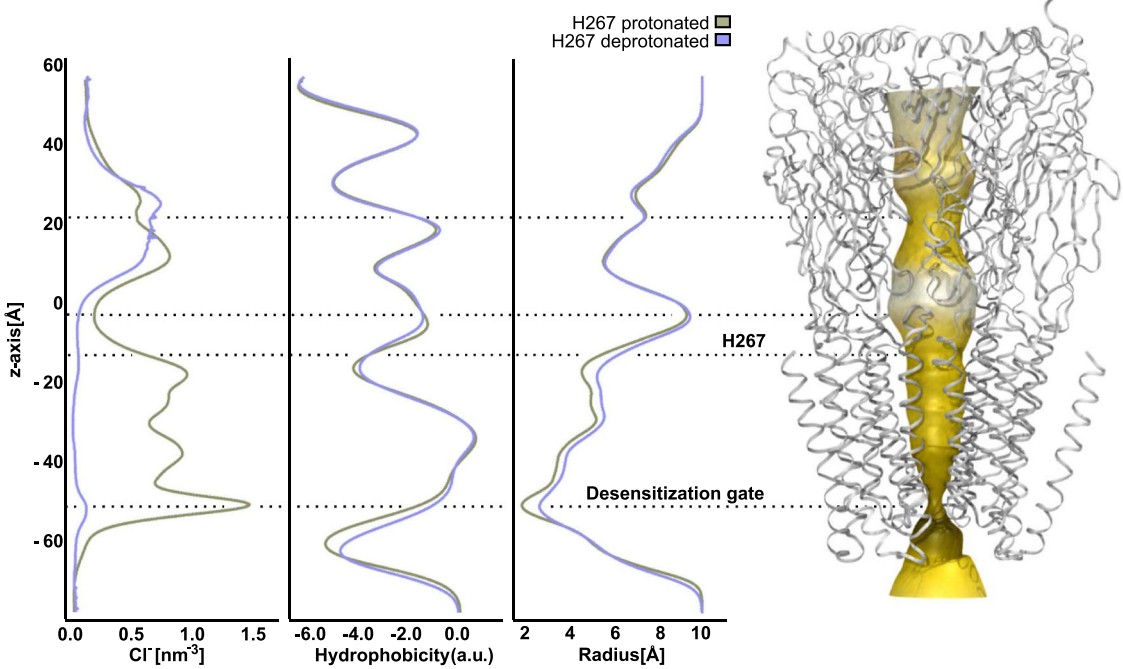

**Fig. 5 Protonation of H267 effects pore features as well as Cl⁻ occupancy.** $Cl^-$ density, hydrophobicity of the pore-facing residues and radius of the pore calculated by the channel annotation programme (CHAP[67]). *Right*: Pore of β3 GABA$_A$R as detected by CHAP. Black dotted lines indicate the projection of the property curves on the pore. Pore properties on the *left* are the averaged over 2 μs MD simulation of β3 GABA$_A$R with H267 protonated and deprotonated β3 GABA$_A$R.

In order to compare the similarity of the three β subunits in the H267 - E270 region, we also examined structures with β1 and β2 subunits. Of these, only heteromeric structures are available[39,40]. Thus, the individual β subunits were superposed for the best fit of the M2 segment. The amino acids of interest are structurally equivalent, and only slightly differ in sidechain rotamers. Thus, we expect the residues H267 and E270 play a structurally equivalent role in β1 and β2 homopentamers (Supplementary Fig. 6).

We propose that interaction between H267 and E270 might play a crucial role in the direct activation of homomeric GABA$_A$ β3 receptors by protons. Mutational studies on the GLIC channel suggests that proton activation occurs also in GLIC allosterically to the orthosteric site, but at the level of multiple loci[41] and we thus cannot exclude that other residues than H267 contribute to proton-induced activation of β3 receptors. Furthermore, additional evidence suggests that other amino acid residues may be involved in proton-induced gating of GLIC where more detailed mechanisms of receptor activation are known[42,43]. There may be a similar picture to that of the GLIC, which has been described to have two electrostatic triads of amino acid residues in the extracellular domain (structurally highly conserved between pLGIC), which are part of a continuous network governing the gating transitions of the receptor. However, due to the absence of open/closed β3 homomeric receptor structures, it is difficult to speculate about the exact determinants and this question remains open for further investigation. The HxxE motif we describe here is unique to the β subunits (Fig. 3c) and absent in GLIC, and is necessary and sufficient for proton sensitivity as demonstrated by the conversion mutants, irrespective of the details of transduction. While MD simulations of GLIC have been conducted[42], a side-by-side comparison with the same method would be needed to get more insight into similarities and differences in the proton interaction sites and transduction mechanisms.

Taken together, our study describes for the first time, that homomeric GABA$_A$ β1, β2 or β3 receptors form proton-gated chloride channels and that appropriate conversion mutants can induce proton gating to ρ1 homopentamers.

β-homomers form readily in recombinant expression systems, their existence in mammalian cells is currently unknown. First evidence for picrotoxin-sensitive $I_{H(\beta x)}$ was obtained in Jurkat T cells (Supplementary Fig. 5a). These cells express β-subunits at much higher abundance than α GABA$_A$ subunits[28] which would be expected to facilitate formation of β homomers. It is tempting to speculate that hyperpolarizing chloride currents upon acidification counterbalances depolarization via ASIC receptors and thus may have a role in immunomodulation[29]. However, this warrants further studies. Different activation states of T-cells might lead to variable expression of proton/ histamine sensitive and GABA-sensitive receptors, respectively[29,31,32].

In studies on GABA-sensitive iPS cell-derived GABANeurons expressing heteropentamers we did not find evidence for $I_{H(\beta x)}$ after 4–6 days in differentiating media (Supplementary Fig. 5b). These data warrant further research as the presence of heteromeric GABA$_A$ receptors in these cells (confirmed by $I_{GABA}$, Supplementary Fig. 5b) may reduce formation of homooligomeric receptors[31,32]. It seems plausible that high expression levels of β-subunits are supportive for formation of β-homomers not only in heterologous expression systems. Given the large diversity of neuronal cell types, broad screens would be needed to potentially detect pH sensitive neuron types.

In fact, evidence for the existence of GABA insensitive receptors formed by β-subunits in neuronal preparations was obtained from studies with alternative agonists such as histamine[5,18,44].

Our results imply that GABA$_A$ β receptors are activated at physiological and pathological pH changes. It is therefore tempting to speculate that such pH sensitive channels formed by ubiquitously expressed subunits in the brain may exist and serve crucial functions under acidotic conditions such as stroke and ischaemia where they can enhance neuroprotective hyperpolarization. Intriguingly, in an experimental model of stroke, upregulation of GABRB3 was connected with a putative

neuroprotective role[45]. Although macroscopic changes in extracellular pH in the brain are tightly controlled, pH fluctuations in specific micro-domains such as the synapse might be relevant[30]. For instance, synaptic vesicles have high proton concentrations (pH 5.7) so vesicle release during high neuronal activity could lower the pH in the synaptic cleft[46].

High levels of RNA of GABA$_A$ beta-subunits are also found in peripheral organs and tissues with a wide range of pH or strongly fluctuating bicarbonate levels such as in endocrine tissues including pancreatic islets [2,47], gastrointestinal tract[48], kidneys[49], respiratory organs[50] as well as in several types of immune cells[29,51] where a local acidification in an inflammatory locus may have relevance for their activation (Supplementary Fig. 5a)[52].

The pH sensitivity of the receptors can serve as a novel tool to analyze the tissue specific expression of homomeric β-receptors in excitable and not excitable cells. To clarify the existence of proton-gated β homomers and their physiological and pathophysiological role in neuronal and/or non-neuronal cells is thus an important goal. In tissues expressing ASIC receptors they might counterbalance a pH dependent depolarisation under acidic conditions (Supplementary Fig. 5). Such receptors (which are already known to be sensitive e.g. to histamine and barbiturates) may form a class of novel potential drug targets.

## Methods

**Animals and animal welfare**. All experiments involving animals were approved by the Austrian Animal Experimentation Ethics Board in compliance with the European convention for the protection of vertebrate animals used for experimental and other scientific purposes ETS no.: 123, which is in line with the EU Directive 2010/63/EU (GZ BMBWF-66.006/0021-V/3b/2019). Every possible effort was taken to minimize the number of animals used.

Female *Xenopus laevis* frogs were purchased from NASCO (Fort Atkinson, USA) and kept in groups in temperature-controlled, continuous-flow water tanks (20 ± 1 °C). The temperature in the holding and testing facilities was fixed to 22 ± 2 °C; the humidity ranged between 40-60%; a 12 h light-dark cycle was in operation (lights on from 07.00 to 19.00).

**Expression of GABA$_A$ receptors in Xenopus laevis oocytes and two-microelectrode voltage-clamp experiments**. Preparation of stage V–VI oocytes from *Xenopus laevis* and synthesis of capped runoff poly(A) cRNA transcripts from linearized cDNA templates (pCMV and pCI vectors) was performed as described elsewhere[53]. Female *Xenopus laevis* frogs were anesthetized by 15 min incubation in a 0.2% MS-222 (methane sulfonate salt of 3-aminobenzoic acid ethyl ester; Sigma Aldrich, Vienna, Austria) solution before removal of parts of the ovaries. Follicle membranes from isolated oocytes were enzymatically digested with 2 mg/mL collagenase (Type 1 A, Sigma Aldrich, Vienna, Austria).

β3 (human) GABA$_A$R and ρ1 (rat) are kindly provided by Petra Scholze, Medical University Vienna. The double α1-β2, α1-β3, γ2-β3 and triple β2-α1-β2, β3-α1-β3, α1-β3-α1 subunit concatemers have been described previously[54].

Mutations β1(S265N), β3(H267A) (rat and human), β3(E270A), and β3(E270Q), β3(H267E-E270H), β3(H267K), β3(H267R) were introduced by site-directed mutagenesis using the QuikChange mutagenesis kit (Agilent Technologies, Vienna, Austria). The coding regions of plasmids were sequenced before experimental use. After cDNA linearization, capped cRNA transcripts were produced using the mMESSAGE mMACHINE® T7 transcription kit (Life Technologies). Selected oocytes were injected with 10–50nL of DEPC-treated water (diethyl pyrocarbonate, Sigma, Vienna, Austria) containing the different cRNAs of GABA$_A$R subunits at a concentration ranging between 200–400 pg/nL/subunit. Commonly cRNAs of α1 and β (β2 or β3) subunits were mixed in a ratio 1:1. For expression of α1β1 receptors cRNAs were, however, mixed in a ratio of 3:1 to reduce formation of β1 homomers. The amount of cRNAs was determined by means of a NanoDrop ND-1000 (Kisker-Biotech, Steinfurt, Germany). Oocytes were stored at +18 °C in ND96 solution (all from Sigma Aldrich, Vienna, Austria).

Chloride currents through heteropentameric GABA$_A$Rs and homomeric subunit receptors were measured at room temperature (20-24 °C) by means of the two-microelectrode voltage clamp technique making use of a TURBO TEC-05X amplifier (npi electronic, Tamm, Germany). $I_{GABA}$ and $I_{H(β)}$ were elicited at a holding potential of -70 mV. Data acquisition was carried out by means of an Axon Digidata 1440 A interface using pCLAMP v.11 (Molecular Devices, Sunnyvale, CA, USA). Functional recordings were made in MES buffer (100 mM NaCl, 3 mM KCl, 1 mM CaCl2, 1 mM MgCl2 and 10 mM MES pH 7.4) equilibrated at the appropriate pHs using 1 M HCl. Microelectrodes were filled with 3 M KCl and had resistances between 1 and 3 MΩ.

**Fast perfusion during voltage clamp experiments**. External solution and drugs were applied by means of a fast perfusion system;[53] drug or control solutions were applied by means of a TECAN Miniprep 60 enabling automation of the experiments (ScreeningTool, npi electronic, Tamm Germany). To elicit chloride currents, the chamber was perfused with GABA- or solutions of different pH respectively, at a volume rate of 200 μL/s[55].

All compounds were dissolved in stock solutions to 100 mM (histamine in ND96, all other compounds in DMSO). Oocytes with maximal current amplitudes >5 μA were discarded to exclude voltage-clamp errors.

Concentration–response curves were generated for different pH and the data were fitted by nonlinear regression analysis using Origin Software (OriginLab Corporation, USA).

Proton-induced currents $I_{H(β)}$ were normalized to the maximum current at pH range 5–4 ($I_{H(β)-max}$) and the concentration–response relationship fitted with the Hill equation:

$$\frac{I_{H(β)}}{I_{H(β)max}} = \frac{1}{1 + \left(\frac{pH_{50}}{pH}\right)^{nH}}$$

where the $pH_{50}$ represents the pH inducing 50% of the maximal current evoked by a saturating concentration and nH is the Hill coefficient. Each data point represents the mean ± SEM from at least 5 oocytes and ≥2 oocyte batches.

**Cell culture and transfection of cells**. Jurkat E 6.1 cells (Sigma Aldrich) were grown at a density of 5×10$^5$ cells/ml in RPMI 1640 medium supplemented with 10% foetal bovine serum (FBS) at 37 °C in a humidified atmosphere at 5% CO2. Every 2 days the cells were split when 80% confluence was reached. CHO cells (Sigma Aldrich) were cultured in DMEM containing 10% heat-inactivated FBS. Cells were grown in culture flasks at 37 °C in a 5% CO2-humidified incubator. Cells were then allowed to grow to 70-80% confluence before transfection. Plasmid coding β3 (human) GABA$_A$R were transfected into CHO cells with TurboFect (ThermoFisher) transfection reagent according to the manufacturer's protocol. Electrophysiological recordings were performed 24–48 h after transfection at room temperature.

**iPS cell-derived iCell GABAergic neurons culture and harvesting**. Frozen stocks of iCell GABANeurons were obtained from CDI containing at least 4 million cells per vial (catalogue number: R1013). Cells were thawed as per the manufacturer's instructions at a density of at least 67,500 cells/cm$^2$ in a 6-well plate. Before planting the cells, the plates were coated with a poly-L-ornithine/laminin coating. In brief 6-well plates were coated with 0.01% poly-L-ornithine (Sigma Aldrich) for 1 h at room temperature; this was then removed, and the dishes were washed twice with sterile distilled water and coated with a 3.3-μg/mL solution of laminin (Sigma Aldrich). The dishes were incubated in a 37 °C incubator for at least 1 h. The laminin was aspirated shortly before addition of the cell suspension. A vial containing at least 4 million iCell GABANeurons was placed in a 37 °C water bath for exactly 3 min. The cells were transferred to a 50-mL Falcon tube, and 1 mL Complete Maintenance Media (containing iCell Neurons medium supplement; CDI) was added slowly (dropwise) to the cells. A further 8 mL Complete Maintenance Media (CDI) was added slowly to the centrifuge tube. Cells were then counted and plated in 6-well plates at a density of 70,000–100,000 cells/cm$^2$. The cells were kept in a 37 °C incubator (5% CO2). After 24 h, media were exchanged for fresh Complete Maintenance Media (100% exchange). Cells could then be kept in a 37 °C incubator (5% CO2) for up to 4 weeks. A 50% media exchange was made every 2–3 days. Cells were cultured for at least 4 days and were typically used on the automated patch-clamp platform 7–10 days after plating. To harvest the cells, media was removed and the iCell GABANeurons were washed with 2 mL phosphate-buffered saline (PBS) containing Ca$^{2+}$/Mg$^{2+}$. After this, prewarmed (37 °C) Accutase (ThermoFisher) was added. The cells were incubated in Accutase for 5 min at 37 °C. The majority of the Accutase was then removed, leaving a thin film over the bottom of the well. Then, 0.5 mL Complete Maintenance Media were added in each well and pipetted up and down to ensure most of the cells were removed from the bottom of the well. An external recording solution (see below) was then added to the cell suspension in a 1:1 ratio before recording.

**Planar patch clamp of mammalian cells**. An automated patch clamp system (SynchroPatch 384, NANION Technologies) for ionic current measurements on suspensions of CHO cells, Jurkat cells or iCell GABANeurons was kindly provided by ChanPharm GmbH (Vienna, Austria). The recording was carried out at room temperature (22–24 ºC) using PatchControl 384 and DataControl 384 (NANION Technologies) software with an external solution containing 140 mM NaCl, 4 mM KCl, 2 mM CaCl2, 1 mM MgCl2, 5 mM glucose, and 10 mM HEPES (pH 7.4 with NaOH). Proton-induced currents were recorded in MES buffer (140 mM NaCl, 4 mM KCl, 2 mM CaCl2, 1 mM MgCl2, 5 mM glucose and 10 mM MES pH 7.4) equilibrated at the appropriate pHs using 1 mM NaOH. The intracellular solution contained 50 mM KCl, 60 mM KF, 10 mM NaCl, 20 mM EGTA, 10 mM HEPES; pH adjusted to 7.2 with KOH. A holding potential of −70 mV was maintained throughout the experiment. Current traces were filtered at 1 or 0.1 kHz. Concentration–response curves were estimated and analysed as described above.

**Statistics and reproducibility**. Statistical significance of differences was calculated using unpaired Student's *t*-test with a level of significance of $p < 0.05$. The data are reported as mean ± SEM. Statistical analysis was performed using Origin Software (OriginLab Corporation, USA). The n numbers in the figure legends signify the number of experiments.

**Chemicals**. All chemicals (γ-Aminobutyric acid, pentobarbital, picrotoxin, amiloride, histamine and bicuculline methiodide) used in this study were obtained from Sigma Aldrich (Vienna, Austria). All compounds were dissolved in stock solutions to 100 mM (histamine and γ-aminobutyric acid in ND96, all other compounds in DMSO).

**Molecular dynamics simulations**. Molecular dynamics (MD) simulations and analyses were performed using Gromacs v.5.1.2[56]. The x-ray structure of the human GABAA β3 homopentamer (4COF[9] was the only structure available when the work started, 7A5V was published later) was inserted into an equilibrated palmitoyloleoyl-phosphatidylcholine (POPC) lipid bilayer membrane using the Charmm-Gui Webserver[57,58], for which Berger lipid parameters were employed[59,60]. For the protein we employed the amber99sb force field[61]. The box was filled with explicit water molecules (SPCE water model[62]) and potassium chloride was used to neutralize the system and add an ion concentration of ~0.18 M. For the ions, corrected monovalent Lennard-Jones parameters for the amber force field was used[63]. The electrostatic interactions cut-off was set to 1.0 nm; long-range electrostatic interactions were treated by the particle-mesh Ewald method[64]. The Lennard-Jones interactions cut-off was set to 1.0 nm. The LINCS algorithm was used to constrain bonds[65]. The simulation temperature was kept constant at 310°K using a V-rescale thermostat; protein and lipids as well as the solvent, together with ions were coupled ($\tau = 0.1$ ps) separately to a temperature bath. Pressure was kept constant at 1 bar using the Parrinello−Rahman barostat algorithm (coupling constant = 2 ps)[66]. Prior to all simulations, we performed steepest descent energy-minimization and equilibrated the system by restraining the position of all heavy atoms with a force constant (fc) of 1000 kJmol-1nm-2 (simulating for 50 ns). An electric field along the channel pore (40 mV.nm-1) was applied. Analysis of the simulations were carried out using the CHAP computational tool[67] as well as the Gromacs analysis tools gmx distance and gmx hbond. The simulation systems were visualized with PyMol[68] and VMD[69].

The beta subunit from 6DW0[39] and 6X3T[40] were utilized to compare the three beta isoforms.

**Reporting summary**. Further information on research design is available in the Nature Research Reporting Summary linked to this article.

## Data availability

Source data underlying figures and MD simulation data that support the findings have been deposited in a Zenodo repository (https://doi.org/10.5281/zenodo.6720229)[70]. Additionally, source data are provided with this paper (Supplementary Data 1). Further data supporting the findings of this publication are available from the corresponding authors upon reasonable request.

## Code availability

The data from the electrophysiological experiments were analysed using the Clampfit 10.1 (Molecular Devices) and DataControl 384 (NANION Technologies) software. Statistical analysis was performed using Origin Software (OriginLab Corporation, USA). Simulation trajectories were collected using the simulation programme GROMACS. Visualization and analysis were performed using VMD, Pymol, Python, and the CHAP computational tool. All of these software packages are publicly available.

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

## Acknowledgements

We thank Eugen Timin for comments on the manuscript. Dinah Weissmann and Christopher Cayzac from ALCEDIAG kindly supplied SH-SY5Y cells. This study was supported by the doctoral programme "Molecular drug targets" W1232 (A.G., T.F., M.S., M.E., A.S.-W., S.H.), the Post-Doc programme "Zukunftskolleg" ZK-81B of the Austrian Science Fund (FWF) (T.F. and E.-M.Z.-P.) and the DOC fellowship 26156 of the Austrian Academy of Sciences (ÖAW) (T.F.). We thank Morten Jensen and Vishwanath Jogini from D. E. Shaw Research for valuable suggestions.

## Author contributions

A.G, E.-M. Z.-P., A.S.-W., S.H. designed research, A.G., T.F., M.S., E.-M.Z.-P., A.W. performed research, A.G., T.F., S.H. analyzed data, A.G., T.F., E.-M.Z.-P., M. E., S.H. wrote the paper.

## Competing interests

The authors declare no competing interests.
