## [Peer Review File · Communications Biology]

Reviewers' comments:

Reviewer #2 (Remarks to the Author):

This is a new study characterizing acidic pH-mediated stimulation of $\beta 3$ subunit in GABARs chloride current in Oocytes expressing the recombinant DNAs. The main findings included reversal of the low pH-mediated channel activation in cells expressing $\beta 3$ H267A mutation, gain of the function in cells expressing $\rho 1$ (G331H-A334E), and simulation analysis of a hydrogen bond formation between H267 and E270, accumulation of Cl^- near the pore structure with protonated H267. Overall, the data are clearly presented, and convincing. The main concerns are that all experiments were performed in non-neuron cells. There is no validation of the findings in physiological conditions or in-depth of discussion about the significance of this acidic pH-mediated regulation mechanism of β subunits in GABA transmission. Therefore, from this point of view, the study is incomplete, and can be further enhanced by additional work.

Reviewer #3 (Remarks to the Author):

This manuscript presents a new claim that GABAA receptors form proton gated chloride channels. The authors also perform an MD simulation capturing a crucial H267-E270 interaction, many details of the said interaction are revealed by the simulation were anticipated by extensive prior mutagenesis studies coupled with functional and biochemical analysis, the direct visualization of these mechanisms is a major advance, and refreshingly the authors do a reasonable job of discussing the MD simulation in context of prior work, although this could be improved. Also, appropriately acknowledged is the fact that it is unclear whether there could be GABA present in trace quantities during the experiment that could impact the major proton gated claim of this manuscript. Also the authors should address the physiological relevance of the pH range tested, as far as my understanding goes the pH even in synapses seldom goes below 6.5. Lastly I am not sure why the authors chose to ignore an array of publications related to a) GABA-A receptor structure for e.g. PMID: 30044221, PMID: 32879488 among others b) GLICS: e.g. PMID: 30541892

A few minor points other than what's already mentioned:

Introduction, para 5, line 2: is Ref 21 the correct citation for the information provided?

Introduction Last para "Mutation H267": please correct the grammar.

Results, Section : Functional properties of proton activated chloride channels, para 2, "Even a small decrease from pH 7.2 to 6.5"- According to me between these two pH values the concentration of H_3O^+ ion changes exponentially as pH is in log scale.

Figure 3c: readers might also benefit from having at least one Gamma subunit in the alignment, I further recommend testing the effects on a tri-heteromeric receptor and comparing the Gamma versus the Rho subunit.

Reviewer #4 (Remarks to the Author):

This is an interesting report showing a novel function of GABA receptor beta subunit, although their existence in vivo has not been shown. The research is well-designed and the results are clear. I think it will be of interest to the people in the field. I want to point out several minor points to consider.

1. A potential significance of the co-expression of beta homomers and heteropentamers in the physiological and pathological conditions should be discussed, although the existence of beta homomers in mammalian cells has not yet been investigated.

2. Figure 2ab: The pH on the figure legend is 5.5. Please make an adequate correction in the manuscript. In addition, the pH used to examine IGABA should be clarified in the manuscript since pH modulates GABA-activated responses on the GABAA receptors.

3. Supplementary figure 1: This figure should be the supplementary figure 2 when arranged in order of appearance. The species of construct should be presented in the figure legend.

4. Supplementary figure 2: which species of beta3 construct was used should be written in the figure legend. This figure should be the supplementary figure 1 when arranged in order of appearance.

Dear reviewers,

We greatly appreciate the time you spent carefully reading and giving us proper feedback on our
manuscript. We have added a substantial new body of data to address the main concerns and corrected
minor typos. We have made changes to several figures and introduced new ones, please find them at the
end of this letter.

**Reviewer #2 (Remarks to the Author):**

This is a new study characterizing acidic pH-mediated stimulation of $\beta 3$ subunit in GABARs chloride
current in Oocytes expressing the recombinant DNAs. The main findings included reversal of the low pH-
mediated channel activation in cells expressing $\beta 3$ H267A mutation, gain of the function in cells
expressing p1 (G331H-A334E), and simulation analysis of a hydrogen bond formation between H267 and
E270, accumulation of Cl⁻ near the pore structure with protonated H267. Overall, the data are clearly
presented, and convincing.

The main concerns are that all experiments were performed in non-neuron cells.

There is no validation of the findings in physiological conditions or in-depth of discussion about the
significance of this acidic pH-mediated regulation mechanism of β subunits in GABA transmission.

Therefore, from this point of view, the study is incomplete, and can be further enhanced by additional
work.

**Point-by-point response to reviewer #2**

***Reviewers comment***

***(1,2) The main concerns are that all experiments were performed in non-neuron cells.***

***There is no validation of the findings in physiological conditions.***

***Response***

To provide some evidence for a potential physiological existence of the novel proton gated
GABA_A receptors built from β subunits, we tested three different cellular systems that are all derived from
cells that express β subunits in their native environment. Specifically, a neuroblastoma cell line (SH-

SY5Y), iPSC-derived GABA neurons, and an immortalized T-cell (Jurkat cells) were tested to consider
both neuron-derived and immune cell derived systems.

In order to be able to compare the current kinetics with the registrations of $I_{H(\beta)}$ on mammalian cells, the
use of planar patch clamps (providing comparable speed of solution exchange as in studies on
mammalian cells) was essential. We made therefore use of a Jurkat cell line that is known to express
GABA_A receptor subunits¹ and can be studied as a suspension with planar patch clamp.

These new data are summarized in an additional Supplementary Figure 5a in Supplementary Information
on p. 6, l. 51 (also please find the figure at the end of this document)

In this cell type rapidly activating and desensitizing currents through ASIC overlaid the currents through β
homomers. ASIC was blocked by amiloride (200 μ M).

The proton-activated current remaining in the presence of amiloride displayed kinetics comparable to
proton activated currents in CHO cells expressing β subunits (Supplementary Figure 2). Their inhibition
by picrotoxin provides further evidence that these currents represent $I_{H(\beta)}$.

A physiological role of the hyperpolarizing I_H could therefore induce (counteract depolarization) inhibition
of the proton-induced depolarization by ASIC and slower desensitization suggest a more long-lasting
effect compared to the transient ASIC currents.

Suspensions of GABA-sensitive iPSC cell-derived GABA neurons were also suitable for planar patch
clamp. These cells expressed heteropentamers (evident from I_{GABA} in Supplementary Figure 5a). We did,
however, not find evidence for $I_{H(\beta)}$ after 4-6 days in differentiating media (Supplementary Figure 5b).

These data warrant further research as the presence of heteromeric GABA_A receptors in these cells
(confirmed by I_{GABA} , Supplementary Figure 5b) may reduce formation of homooligomeric receptors^{2,3}. In
the revised MS we discuss, however, that high expression levels of β subunits is supportive for formation
of β -homomers not only in heterologous expression systems. Given the large diversity of neuronal cell
types, broad screens would be needed to potentially detect pH sensitive neuron types.

In a third series of experiments neuroblastoma cells (SH-SY5Y) were detached and suspensions of these
cells studied with planar patch clamp. Experiments were performed on non-differentiated cells as well as
on differentiated cells where retinoic acid was used (in this case we carried out the experiments on day
10 after the start of differentiation). We observed neither I_{GABA} nor evidence for $I_{H(\beta)}$.

The unequivocal identification of $I_{H(\beta)}$ in native tissues remains, thus, a major challenge for future studies.
This is also complicated by the fact that no specific inhibitors for β homomers are currently available.
We would, however, like to emphasize that β pentamers have received considerable attention as putative
histamine gated anion channels^{4,5}.

**Reviewers comment**

**(3) in-depth of discussion about the significance of this acidic pH-mediated regulation mechanism**
**of β subunits in GABA transmission**

**Response**

According to the reviewer's suggestion we discuss the possible role of pH mediated regulation
mechanism of β -subunits in more detail.

See p. 11, l. 270-307 for a more extensive discussion on the possible expression of receptors and their
relevance:

" β -homomers form readily in recombinant expression systems, their existence in mammalian cells is
currently unknown. First evidence for picrotoxin-sensitive $I_{H(\beta x)}$ was obtained in Jurkat T cells
(Supplementary Figure 5a). These cells express β -subunits at much higher abundance than α GABA_A
subunits²⁸ which would be expected to facilitate formation of β homomers. It is tempting to speculate that
hyperpolarizing chloride currents upon acidification counterbalances depolarization via ASIC receptors
and thus may have a role in immunomodulation²⁹. However, this warrants further studies. Different
activation states of T-cells might lead to variable expression of proton/ histamine sensitive and GABA
sensitive receptors, respectively^{29,31,32}.

In studies on GABA-sensitive iPS cell-derived GABA neurons expressing heteropentamers we did not find
evidence for $I_{H(\beta x)}$ after 4-6 days in differentiating media (Supplementary Figure 5b). These data warrant
further research as the presence of heteromeric GABA_A receptors in these cells (confirmed by I_{GABA_A} ,
Supplementary Figure 5b) may reduce formation of homooligomeric receptors^{31,32}. It seems plausible that
high expression levels of β -subunits are supportive for formation of β -homomers not only in heterologous
expression systems. Given the large diversity of neuronal cell types, broad screens would be needed to
potentially detect pH sensitive neuron types.

In fact, evidence for the existence of GABA insensitive receptors formed by β -subunits in neuronal
preparations was obtained from studies with alternative agonists such as histamine^{5,18,44}.

Our results imply that GABA_A β receptors are activated at physiological and pathological pH changes. It
is therefore tempting to speculate that such pH sensitive channels formed by ubiquitously expressed
subunits in the brain may exist and serve crucial functions under acidotic conditions such as stroke and
ischemia where they can enhance neuroprotective hyperpolarization. Intriguingly, in an experimental
model of stroke, upregulation of GABRB3 was connected with a putative neuroprotective role⁴⁵. Although
macroscopic changes in extracellular pH in the brain are tightly controlled, pH fluctuations in specific
micro-domains such as the synapse might be relevant³⁰. For instance, synaptic vesicles have high proton
concentrations (pH 5.7) so vesicle release during high neuronal activity could lower the pH in the synaptic
cleft⁴⁶.

High levels of RNA of GABA_A beta-subunits are also found in peripheral organs and tissues with a wide
range of pH or strongly fluctuating bicarbonate levels such as in endocrine tissues including pancreatic
islets^{2,47}, gastrointestinal tract⁴⁸, kidneys⁴⁹, respiratory organs⁵⁰ as well as in several types of immune
cells^{29,51} where a local acidification in an inflammatory locus may have relevance for their activation
(Supplementary Figure 5a)⁵².

The pH sensitivity of the receptors can serve as a novel tool to analyze the tissue specific expression of
homomeric β -receptors in excitable and not excitable cells. To clarify the existence of proton-gated β
homomers and their physiological and pathophysiological role in neuronal and/or non-neuronal cells is
thus an important goal. In tissues expressing ASIC receptors they might counterbalance a pH dependent
depolarisation under acidic conditions (Supplementary Figure 5). Such receptors (which are already
known to be sensitive e.g. to histamine and barbiturates) may form a class of novel potential drug
targets.”

**Reviewer #3 (Remarks to the Author):**

This manuscript presents a new claim that GABAA receptors form proton gated chloride channels.

The authors also perform an MD simulation capturing a crucial H267-E270 interaction, many details of the
said interaction are revealed by the simulation were anticipated by extensive prior mutagenesis studies
coupled with functional and biochemical analysis, the direct visualization of these mechanisms is a major
advance, and refreshingly the authors do a reasonable job of discussing the MD simulation in context of
prior work, although this could be improved.

Also, appropriately acknowledged is the fact that it is unclear whether there could be GABA present in
trace quantities during the experiment that could impact the major proton gated claim of this manuscript.

Also the authors should address the physiological relevance of the pH range tested, as far as my
understanding goes the pH even in synapses seldom goes below 6.5.

Lastly I am not sure why the authors chose to ignore an array of publications related to

a) GABA-A receptor structure for e.g. PMID:30044221, PMID: 32879488 among others

b) GLICS: e.g. PMID: 30541892

A few minor points other than what's already mentioned:

Introduction, para 5, line 2: is Ref 21 the correct citation for the information provided?

Introduction Last para "Mutation H267": please correct the grammar.

Results, Section : Functional properties of proton activated chloride channels, para 2, "Even a small
decrease from pH 7.2 to 6.5"- According to me between these two pH values the concentration of H₃O⁺
ion changes exponentially as pH is in log scale.

Figure 3c: readers might also benefit from having at least one Gamma subunit in the alignment, I further
recommend testing the effects on a tri-heteromeric receptor and comparing the Gamma versus the Rho
subunit.

**Point-by-point response to reviewer #3**

***Reviewers comment***

***(1) “refreshingly the authors do a reasonable job of discussing the MD simulation in context of***
***prior work, although this could be improved”***

***Response***

According to the reviewers comment we have extended the discussion on MD simulations in the context
of what is already known from prior work on the GLIC receptor and added a new paragraph on p.10, l.
249-264

“We propose that interaction between H267 and E270 might play a crucial role in the direct activation of
homomeric GABA_A β 3 receptors by protons. Mutational studies on the GLIC channel suggests that proton
activation occurs also in GLIC allosterically to the orthosteric site, but at the level of multiple loci⁴¹ and we
thus cannot exclude that other residues than H267 contribute to proton-induced activation of β 3
receptors. Furthermore, additional evidence suggests that other amino acid residues may be involved in
proton-induced gating of GLIC where more detailed mechanisms of receptor activation are known^{42,43}.
There may be a similar picture to that of the GLIC, which has been described to have two electrostatic
triads of amino acid residues in the extracellular domain (structurally highly conserved between pLGIC),
which are part of a continuous network governing the gating transitions of the receptor. However, due to
the absence of open/closed β 3 homomeric receptor structures, it is difficult to speculate about the exact
determinants and this question remains open for further investigation. The HxxE motif we describe here is
unique to the β subunits (Fig. 3c) and absent in GLIC, and is necessary and sufficient for proton
sensitivity as demonstrated by the conversion mutants, irrespective of the details of transduction. While
MD simulations of GLIC have been conducted⁴², a side-by-side comparison with the same method would
be needed to get more insight into similarities and differences in the proton interaction sites and
transduction mechanisms.”

***Reviewers comment***

***(2) “Also, appropriately acknowledged is the fact that it is unclear whether there could be GABA***
***present in trace quantities during the experiment that could impact the major proton gated claim***
***of this manuscript”***

**Response**

No traces of GABA were present in the experiments shown in Fig. 1, Fig. 2, Fig.3 and Supplementary
Figure 2, Supplementary Figure 3 as there was no GABA added to the test solutions. We can completely
exclude the presence of any traces of GABA.

**Reviewers comment**

**(3) “Also the authors should address the physiological relevance of the pH range tested, as far as**
**my understanding goes the pH even in synapses seldom goes below 6.5”**

**Response**

This question was addressed in Fig. 2 where we illustrate $I_{H(\beta)}$ evoked by pH changes from 7.2 to 6.5.

Furthermore, physiological and pathophysiological pH changes are now discussed in more detail on p.
11, l. 270-307.

“ β -homomers form readily in recombinant expression systems, their existence in mammalian cells is
currently unknown. First evidence for picrotoxin-sensitive $I_{H(\beta_x)}$ was obtained in Jurkat T cells
(Supplementary Figure 5a). These cells express β -subunits at much higher abundance than α GABA_A
subunits²⁸ which would be expected to facilitate formation of β homomers. It is tempting to speculate that
hyperpolarizing chloride currents upon acidification counterbalances depolarization via ASIC receptors
and thus may have a role in immunomodulation²⁹. However, this warrants further studies. Different
activation states of T-cells might lead to variable expression of proton/ histamine sensitive and GABA
sensitive receptors, respectively^{29,31,32}.

In studies on GABA-sensitive iPS cell-derived GABA neurons expressing heteropentamers we did not find
evidence for $I_{H(\beta_x)}$ after 4-6 days in differentiating media (Supplementary Figure 5b). These data warrant
further research as the presence of heteromeric GABA_A receptors in these cells (confirmed by I_{GABA} ,
Supplementary Figure 5b) may reduce formation of homooligomeric receptors^{31,32}. It seems plausible that
high expression levels of β -subunits are supportive for formation of β -homomers not only in heterologous
expression systems. Given the large diversity of neuronal cell types, broad screens would be needed to
potentially detect pH sensitive neuron types.

In fact, evidence for the existence of GABA insensitive receptors formed by β -subunits in neuronal
preparations was obtained from studies with alternative agonists such as histamine^{5,18,44}.

Our results imply that GABA_A β receptors are activated at physiological and pathological pH changes. It
is therefore tempting to speculate that such pH sensitive channels formed by ubiquitously expressed
subunits in the brain may exist and serve crucial functions under acidotic conditions such as stroke and
ischemia where they can enhance neuroprotective hyperpolarization. Intriguingly, in an experimental
model of stroke, upregulation of GABRB3 was connected with a putative neuroprotective role⁴⁵. **Although**
**macroscopic changes in extracellular pH in the brain are tightly controlled, pH fluctuations in specific**
**micro-domains such as the synapse might be relevant³⁰. For instance, synaptic vesicles have high proton**
**concentrations (pH 5.7) so vesicle release during high neuronal activity could lower the pH in the synaptic**
**cleft⁴⁶.**

High levels of RNA of GABA_A beta-subunits are also found in peripheral organs and tissues with a wide
range of pH or strongly fluctuating bicarbonate levels such as in endocrine tissues including pancreatic
islets^{2,47}, gastrointestinal tract⁴⁸, kidneys⁴⁹, respiratory organs⁵⁰ as well as in several types of immune
cells^{29,51} where a local acidification in an inflammatory locus may have relevance for their activation
(Supplementary Figure 5a)⁵².

The pH sensitivity of the receptors can serve as a novel tool to analyze the tissue specific expression of
homomeric β-receptors in excitable and not excitable cells. To clarify the existence of proton-gated β
homomers and their physiological and pathophysiological role in neuronal and/or non-neuronal cells is
thus an important goal. In tissues expressing ASIC receptors they might counterbalance a pH dependent
depolarisation under acidic conditions (Supplementary Figure 5). Such receptors (which are already
known to be sensitive e.g. to histamine and barbiturates) may form a class of novel potential drug
targets.”

***Reviewers comment***

**(4) Lastly I am not sure why the authors chose to ignore *an array of publications related to***

***a) GABA-A receptor structure for e.g. PMID:30044221, PMID: 32879488 among others***

***b) GLICS: e.g. PMID: 30541892***

***Response***

All references mentioned by the reviewer have been cited and discussed in the revised version of the
Manuscript and Supplementary Information (Supplementary Figure 6, also see Supplementary Figure 6 in
the end of the response file).

The references related to beta subunit structures are included in Supplementary Information, more
specifically to align beta subunits, the structure of $\beta 1$ subunit (6DW0) from the *PMID:30044221*, of $\beta 2$
subunits (6X3T) from *PMID: 32879488* was used alongside with two $\beta 3$ subunits (4COF and 7A5V).

Their discussion (and additional discussion on GLIC touching on the *PMID: 30541892* paper) are now
included into the new paragraphs on p. 10, l. 243-264.

“In order to compare the similarity of the three β subunits in the H267 - E270 region, we also examined
structures with $\beta 1$ and $\beta 2$ subunits. Of these, only heteromeric structures are available^{39,40}. Thus, the
individual β subunits were superposed for the best fit of the M2 segment. The amino acids of interest are
structurally equivalent, and only slightly differ in sidechain rotamers. Thus, we expect the residues H267
and E270 play a structurally equivalent role in $\beta 1$ and $\beta 2$ homopentamers (Supplementary Figure 6).

We propose that interaction between H267 and E270 might play a crucial role in the direct activation of
homomeric GABA_A $\beta 3$ receptors by protons. Mutational studies on the GLIC channel suggests that proton
activation occurs also in GLIC allosterically to the orthosteric site, but at the level of multiple loci⁴¹ and we
thus cannot exclude that other residues than H267 contribute to proton-induced activation of $\beta 3$
receptors. Furthermore, additional evidence suggests that other amino acid residues may be involved in
proton-induced gating of GLIC where more detailed mechanisms of receptor activation are known^{42,43}.
There may be a similar picture to that of the GLIC, which has been described to have two electrostatic
triads of amino acid residues in the extracellular domain (structurally highly conserved between pLGIC),
which are part of a continuous network governing the gating transitions of the receptor. However, due to
the absence of open/closed $\beta 3$ homomeric receptor structures, it is difficult to speculate about the exact
determinants and this question remains open for further investigation. The HxxE motif we describe here is
unique to the β subunits (Fig. 3c) and absent in GLIC, and is necessary and sufficient for proton
sensitivity as demonstrated by the conversion mutants, irrespective of the details of transduction. While
MD simulations of GLIC have been conducted⁴², a side-by-side comparison with the same method would
be needed to get more insight into similarities and differences in the proton interaction sites and
transduction mechanisms.”

**Reviewers comment**

**(5) Introduction, para 5, line 2: is Ref 21 the correct citation for the information provided?**

**Response**

We have checked that the reference is correct. In addition, we added another source (now references 24
and 25).

**Reviewers comment**

**(6) Introduction Last para “Mutation H267”: please correct the grammar.**

**Response**

The grammar has been corrected, see p.4, l 58

“The mutation H267A completely abolished proton sensitivity”

**Reviewers comment**

**(7) Results, Section : Functional properties of proton activated chloride channels, para 2, “Even a**
**small decrease from pH 7.2 to 6.5”- According to me between these two pH values the**
**concentration of H₃O⁺ ion changes exponentially as pH is in log scale**

**Response**

Indeed, the word “small” was misleading as change of pH by 0.7 (from 7.2 to 6.5) corresponds to a 5-fold
increase of the H⁺ concentration { $10^{0.7} \approx 5$ }. Correspondingly we removed the word small and write on p.
5, l. 105: “A decrease from pH 7.2 to 6.5 ...”.

**Reviewers comment**

**(8) Figure 3c: readers might also benefit from having at least one Gamma subunit in the alignment**

**Response**

According to the reviewer’s suggestion we have included a γ subunit into Fig. 3c. See the completed
Figure 3c on p. 27, l. 637 (also see the Fig. 3 at the end of this document).

**Reviewers comment**

**(9) I further recommend testing the effects on a tri-heteromeric receptor and comparing the**
**Gamma versus the Rho subunit.**

**Response**

According to the reviewer's suggestion we have performed these additional experiments with tri-
heteromeric receptor ($\alpha1-\beta3-\alpha1/\gamma2-\beta3$) and included the data into the Manuscript (p.5, l.113).

"Subunit concatenation ($\alpha1-\beta2/\beta2-\alpha1-\beta2$, $\alpha1-\beta3/\beta3-\alpha1-\beta3$ and $\alpha1-\beta3-\alpha1/\gamma2-\beta3$) prevented both the
formation of β homomers and at the same time activation of $I_{H(\beta)}$ (Supplementary Figure 4)."

See additional Supplementary Figure 4c in SI on p. 5, l. 46 (also see the Supplementary Figure 4 at the
end of this document).

**Reviewer #4 (Remarks to the Author):**

This is an interesting report showing a novel function of GABA receptor beta subunit, although their
existence in vivo has not been shown. The research is well-designed and the results are clear. I think it
will be of interest to the people in the field. I want to point out several minor points to consider.

1. A potential significance of the co-expression of beta homomers and heteropentamers in the
physiological and pathological conditions should be discussed, although the existence of beta homomers
in mammalian cells has not yet been investigated.

2. Figure 2ab: The pH on the figure legend is 5.5. Please make an adequate correction in the manuscript.
In addition, the pH used to examine I_{GABA} should be clarified in the manuscript since pH modulates GABA-
activated responses on the GABAA receptors.

3. Supplementary figure 1: This figure should be the supplementary figure 2 when arranged in order of
appearance. The species of construct should be presented in the figure legend.

4. Supplementary figure 2: which species of beta3 construct was used should be written in the figure
legend. This figure should be the supplementary figure 1 when arranged in order of appearance.

**Point-by-point response to reviewer #4**

***Reviewers comment***

**(1) A potential significance of the co-expression of beta homomers and heteropentamers in the**
**physiological and pathological conditions should be discussed, although the existence of beta**
**homomers in mammalian cells has not yet been investigated.**

***Response***

In order to address the reviewer's suggestion, we have performed additional experiments on Jurkat cells
expressing predominantly GABA_A receptor β -subunits and iPS cell-derived GABANeurons expressing
heteropentamers. Jurkat cells grow as a suspension and GABANeurons can be transferred to a
suspension after detaching from the substrate. Both cell types were therefore suitable for studies using
the planar patch clamp technique.

These new data are summarized in an additional Supplementary Figure 5 on p. 6, l. 51 (also see the
Supplementary Figure 5 at the end of this document).

.

See p. 11, l. 270-307 for a more thorough discussion on the possible expression of receptors and their
relevance.

“ β -homomers form readily in recombinant expression systems, their existence in mammalian cells is
currently unknown. First evidence for picrotoxin-sensitive $I_{H(\beta x)}$ was obtained in Jurkat T cells
(Supplementary Figure 5a). These cells express β -subunits at much higher abundance than α $GABA_A$
subunits²⁸ which would be expected to facilitate formation of β homomers. It is tempting to speculate that
hyperpolarizing chloride currents upon acidification counterbalances depolarization via ASIC receptors
and thus may have a role in immunomodulation²⁹. However, this warrants further studies. Different
activation states of T-cells might lead to variable expression of proton/ histamine sensitive and GABA
sensitive receptors, respectively^{29,31,32}.

In studies on GABA-sensitive iPS cell-derived GABA neurons expressing heteropentamers we did not find
evidence for $I_{H(\beta x)}$ after 4-6 days in differentiating media (Supplementary Figure 5b). These data warrant
further research as the presence of heteromeric $GABA_A$ receptors in these cells (confirmed by I_{GABA} ,
Supplementary Figure 5b) may reduce formation of homooligomeric receptors^{31,32}. It seems plausible that
high expression levels of β -subunits are supportive for formation of β -homomers not only in heterologous
expression systems. Given the large diversity of neuronal cell types, broad screens would be needed to
potentially detect pH sensitive neuron types.

In fact, evidence for the existence of GABA insensitive receptors formed by β -subunits in neuronal
preparations was obtained from studies with alternative agonists such as histamine^{5,18,44}.

Our results imply that $GABA_A$ β receptors are activated at physiological and pathological pH changes. It
is therefore tempting to speculate that such pH sensitive channels formed by ubiquitously expressed
subunits in the brain may exist and serve crucial functions under acidotic conditions such as stroke and
ischemia where they can enhance neuroprotective hyperpolarization. Intriguingly, in an experimental
model of stroke, upregulation of GABRB3 was connected with a putative neuroprotective role⁴⁵. Although
macroscopic changes in extracellular pH in the brain are tightly controlled, pH fluctuations in specific
micro-domains such as the synapse might be relevant³⁰. For instance, synaptic vesicles have high proton
concentrations (pH 5.7) so vesicle release during high neuronal activity could lower the pH in the synaptic
cleft⁴⁶.

High levels of RNA of GABA_A beta-subunits are also found in peripheral organs and tissues with a wide
range of pH or strongly fluctuating bicarbonate levels such as in endocrine tissues including pancreatic
islets^{2,47}, gastrointestinal tract⁴⁸, kidneys⁴⁹, respiratory organs⁵⁰ as well as in several types of immune
cells^{29,51} where a local acidification in an inflammatory locus may have relevance for their activation
(Supplementary Figure 5a)⁵².

The pH sensitivity of the receptors can serve as a novel tool to analyze the tissue specific expression of
homomeric β -receptors in excitable and not excitable cells. To clarify the existence of proton-gated β
homomers and their physiological and pathophysiological role in neuronal and/or non-neuronal cells is
thus an important goal. In tissues expressing ASIC receptors they might counterbalance a pH dependent
depolarisation under acidic conditions (Supplementary Figure 5). Such receptors (which are already
known to be sensitive e.g. to histamine and barbiturates) may form a class of novel potential drug
targets.”

***Reviewers comment***

**(2) Figure 2ab: The pH on the figure legend is 5.5. Please make an adequate correction in the**
**manuscript. In addition, the pH used to examine I_{GABA} should be clarified in the manuscript since**
**pH modulates GABA-activated responses on the GABA_A receptors.**

***Response***

We have made an appropriate correction of the figure legend, included the pH in the subscript, see Fig. 2
on p. 25, l. 617, l.619 and l. 622.

All recordings of I_{GABA} were made at physiological pH (pH= 7.2). This is now clarified in the subscript of
Fig.2 p.25, l.620, Fig.4 p.29, l.651 and Supplementary Figure 4 p.5, l.50

We also specify the pH values of the solution used to induce I_{GABA} on p.5, l.97

***Reviewers comment***

**(3) Supplementary figure 1: This figure should be the supplementary figure 2 when arranged in**
**order of appearance. The species of construct should be presented in the figure legend.**

***Response***

We changed the order of the figures according to appearance in the text and specified the species in the
legend, see SI p.3

***Reviewers comment***

**(4) Supplementary figure 2: which species of beta3 construct was used should be written in the**
**figure legend. This figure should be the supplementary figure 1 when arranged in order of**
**appearance.**

***Response***

According to the reviewer's suggestion we specified the species of beta3 construct in the subscript and
changed the order (swap Supplementary Figure 2 with Supplementary Figure 1), see SI p.2 and p.3

- 1. Dionisio, L., Arias, V., Bouzat, C. & Esandi, M. del C. GABAA receptor plasticity in Jurkat T cells.
*Biochimie* **95**, 2376–2384 (2013).
- 2. Sieghart, W. & Sperk, G. Subunit composition, distribution and function of GABA(A) receptor subtypes.
*Curr Top Med Chem* **2**, 795–816 (2002).
- 3. Angelotti, T. P. & Macdonald, R. L. Assembly of GABAA receptor subunits: alpha 1 beta 1 and alpha 1
beta 1 gamma 2S subunits produce unique ion channels with dissimilar single- channel properties. *J.*
*Neurosci.* **13**, 1429–1440 (1993).
- 4. Fleck, M. W., Thomson, J. L. & Hough, L. B. Histamine-gated ion channels in mammals? *Biochem*
*Pharmacol* **83**, 1127–1135 (2012).
- 5. Nakane, T. *et al.* Single-particle cryo-EM at atomic resolution. *Nature* **587**, 152–156 (2020).

Figures (that have been added or improved in the manuscript)

Figure 2: the correct pH values in the subscript have been indicated; the pH value for I_{GABA} was specified

Fig. 2: Functional properties of proton activated homomeric β receptors.

a Activation kinetics of $I_{H(\beta_3)}$ (pH 5) through human homomeric β_3 receptors and I_{GABA} (GABA 100 μ M, pH
7.2) through $\alpha 1\beta_3$ GABA_A receptors at saturating agonist concentrations. The current traces illustrate
slower activation of $I_{H(\beta_3)}$ ($t_{peak} = 1.91$ s) compared to I_{GABA} ($t_{peak} = 0.68$ s). b Desensitisation kinetics of $I_{H(\beta_3)}$
and I_{GABA} during long lasting exposure to pH 5 or to GABA 100 μ M. Desensitization half time for $I_{H(\beta_3)}$ is

$t_{0.5} = 58.5$ s compared to $t_{0.5} = 33.6$ s of I_{GABA} . pH in **(a, b)** was changed from 9 to 5. **c** Representative $I_{H(\beta 3)}$
induced by switching the pH from 7.2 to 6.5. **d** The pH-dependence of steady-state desensitization and of
activation of receptors composed of the indicated β -subunits. Oocytes were conditioned for 10 minutes at
the indicated pH, and receptors were subsequently activated by shifting the pH rapidly to 4.5 in order to
estimate the the fraction of available (not yet desensitized) channels. The normalized current response is
plotted as a function of the conditioning pH (superimposed data for current activation (dash lines) are
taken from Fig. 1). Half-maximal desensitisation occurs at pH 6.4 ± 0.02 for $\beta 1(S265N)$, at pH 6.97 ± 0.06
for $\beta 2$ and at pH 6.6 ± 0.15 ($n = 4$) for $\beta 3$ homomers. pH was reduced from 9 to the indicated values. The
data are presented as mean values \pm SEM, $n=5$.

**Figure 3:** $\gamma 2$ subunit was included in the alignment (Fig.3c)

**Fig. 3: Loss of function ($\beta 3(\text{H267A})$) mutation in GABA_A subunit homomers and a hydrogen bond**
 **formation between H267 and E270.**

**a** Location of H267 in the pore region of the crystal structure of $\beta 3$ homomers (the position of H267 is
 indicated with violet spheres)⁹. **b** Mutation H267A in the human subunits $\beta 3$ subunit prevented activation
 of $I_{H(\beta 3(\text{H267A}))}$ but, instead, induced pH-dependent inhibition of large baseline currents (upper traces).

Dotted line illustrates zero current level. Lower traces illustrate retained sensitivity for other agonists (100
μM pentobarbital, 1mM histamine and 100 μM bicuculline). **c** Sequence alignment of the transmembrane
M2 α -helix of $\alpha 1$, $\beta 1$ -3 (human), $\gamma 2$ (human) and $\rho 1$ (rat) subunits (segment does not differ between rat
and human). The location of H267 and E270 in β subunits and the corresponding homologous positions
in the $\rho 1$ subunit (double mutant $\rho 1(\text{G331H-A334E})$) are illustrated. **d** Histogram of the number of
hydrogen bonds between protonated and deprotonated H267 and E270. **e** Snapshots of the ring structure
observed in the MD runs.

**Supplementary Figure 4:** additional experiments with the tri-heteromeric receptor ($\alpha 1\text{-}\beta 3\text{-}\alpha 1/\gamma 2\text{-}\beta 3$) were
performed, the results are included in the Supplementary Figure 4c.

**Supplementary Figure 4. Concatenation of subunits prevents activation of $I_{H(\beta)}$.** GABA_A receptors
composed of concatenated subunits $\alpha 1\text{-}\beta 2/\beta 2\text{-}\alpha 1\text{-}\beta 2$ (rat) (a), $\alpha 1\text{-}\beta 3/\beta 3\text{-}\alpha 1\text{-}\beta 3$ (rat) (b) and $\alpha 1\text{-}\beta 3\text{-}\alpha 1/\gamma 2\text{-}\beta 3$
(rat) (c) are not activated by changing the pH from 9 to indicated values (a, b and c, respectively) but
activated by GABA (pH 7.2).

**Supplementary Figure 5. Proton- and GABA-induced currents in Jurkat cells (a) and iPS cell-**
**derived iCell GABAergic neurons (b).**

**a** Representative currents recorded from Jurkat cells upon shifting the pH from 7.2 to 5 alone (left) or in
the presence of 200 μM amiloride (middle overlaid currents, blue). Application of pH 5 produced ASIC-like
fast activating currents, whereas co-application of pH 5 with amiloride (an ASIC blocker²) resulted in
slower activating and desensitising currents that were completely blocked by picrotoxin (100 μM).
Application of GABA did not activate I_{GABA} in these cells. Average mean current amplitudes are $440 \pm$
118 pA for pH 5-induced currents and 34 ± 4 pA for pH 5 in presence of 200 μM amiloride ($n = 11$, the
data are presented as mean values \pm SEM).

**b** Representative currents recorded from iPS cell-derived iCell GABAergic neurons cells. Application of
pH 5 resulted in a current (left) which was completely abolished by co-application with amiloride (middle
trace), consistent with an ASIC current. I_{GABA} (induced by 100 μM GABA) is shown on the right (indicating
the presence of heteromeric GABA_A receptors in these neurons). Average current amplitudes are $225 \pm$
44 pA for pH 5-induced ASIC currents and 308 ± 35 pA for GABA-induced currents ($n = 5$, the data are
presented as mean values \pm SEM).

**Supplementary Figure 6. Alignment of the M2 segment of different experimental structures of beta**
**subunits.** The protein is represented as cartoon; residues H267 and E270 (numbering corresponds to β 3
numbers) are shown as sticks. The β 3 subunits are colored in slate blue (4COF³) and dark blue (7A5V⁴),
β 1 subunit in grey (6DW0⁵) and the β 2 subunits of the cryo-EM structure 6X3T⁶ in light orange (chain A)
and dark orange (chain E). Positions of H267 and E270 are equivalent and only slight differences in the
rotameric state of are visible. We therefore expect these residues play a structurally equivalent role in
homopentamers of β 1 and β 2. In the heteropentamers, the TMD adopts a different conformation, which
limits this structural comparison due to the lacking structure of homomeric β 1 or β 2 structures.

REVIEWERS' COMMENTS:

Reviewer #2 (Remarks to the Author):

Previous concerns have been addressed.

Reviewer #3 (Remarks to the Author):

I agree with the authors addition of new body of data, and it addresses any concerns that I had. The manuscript is in a much better shape and provides a more comprehensive view of the question at hand. Please accept my congratulations on this wonderful piece of work.

Reviewer #4 (Remarks to the Author):

The authors have addressed all of my comments. This manuscript is now suitable for publication.

Dear reviewers,

We greatly appreciate the time you spent carefully reading and giving us proper feedback on our manuscript. We have added a substantial new body of data to address the main concerns and corrected minor typos. We have made changes to several figures and introduced new ones, please find them at the end of this letter.

Reviewer #2 (Remarks to the Author):

This is a new study characterizing acidic pH-mediated stimulation of $\beta 3$ subunit in GABARs chloride current in Oocytes expressing the recombinant DNAs. The main findings included reversal of the low pH-mediated channel activation in cells expressing $\beta 3$ H267A mutation, gain of the function in cells expressing p1 (G331H-A334E), and simulation analysis of a hydrogen bond formation between H267 and E270, accumulation of Cl⁻ near the pore structure with protonated H267. Overall, the data are clearly presented, and convincing.

The main concerns are that all experiments were performed in non-neuron cells.

There is no validation of the findings in physiological conditions or in-depth of discussion about the significance of this acidic pH-mediated regulation mechanism of β subunits in GABA transmission.

Therefore, from this point of view, the study is incomplete, and can be further enhanced by additional work.

Point-by-point response to reviewer #2

Reviewers comment

(1,2) The main concerns are that all experiments were performed in non-neuron cells.

There is ***no validation of the findings in physiological conditions.***

Response

To provide some evidence for a potential physiological existence of the novel proton gated GABA_A receptors built from β subunits, we tested three different cellular systems that are all derived from cells that express β subunits in their native environment. Specifically, a neuroblastoma cell line (SH-

SY5Y), iPSC-derived GABA neurons, and an immortalized T-cell (Jurkat cells) were tested to consider both neuron-derived and immune cell derived systems.

In order to be able to compare the current kinetics with the registrations of $I_{H(\beta)}$ on mammalian cells, the use of planar patch clamps (providing comparable speed of solution exchange as in studies on mammalian cells) was essential. We made therefore use of a Jurkat cell line that is known to express GABA_A receptor subunits¹ and can be studied as a suspension with planar patch clamp.

These new data are summarized in an additional Supplementary Figure 5a in Supplementary Information on p. 6, l. 51 (also please find the figure at the end of this document)

In this cell type rapidly activating and desensitizing currents through ASIC overlaid the currents through β homomers. ASIC was blocked by amiloride (200 μ M).

The proton-activated current remaining in the presence of amiloride displayed kinetics comparable to proton activated currents in CHO cells expressing β subunits (Supplementary Figure 2). There inhibition by picrotoxin provides further evidence that these currents represent $I_{H(\beta)}$.

A physiological role of the hyperpolarizing I_H could therefore induce (counteract depolarization) inhibition of the proton-induced depolarization by ASIC and slower desensitization suggest a more long-lasting effect compared to the transient ASIC currents.

Suspensions of GABA-sensitive iPSC cell-derived GABA neurons were also suitable for planar patch clamp. These cells expressed heteropentamers (evident from I_{GABA} in Supplementary Figure 5a). We did, however, not find evidence for $I_{H(\beta x)}$ after 4-6 days in differentiating media (Supplementary Figure 5b). These data warrant further research as the presence of heteromeric GABA_A receptors in these cells (confirmed by I_{GABA} , Supplementary Figure 5b) may reduce formation of homooligomeric receptors^{2,3}. In the revised MS we discuss, however, that high expression levels of β subunits is supportive for formation of β -homomers not only in heterologous expression systems. Given the large diversity of neuronal cell types, broad screens would be needed to potentially detect pH sensitive neuron types.

In a third series of experiments neuroblastoma cells (SH-SY5Y) were detached and suspensions of these cells studied with planar patch clamp. Experiments were performed on non-differentiated cells as well as on differentiated cells where retinoic acid was used (in this case we carried out the experiments on day 10 after the start of differentiation). We observed neither I_{GABA} nor evidence for $I_{H(\beta x)}$.

The unequivocal identification of $I_{H(\beta)}$ in native tissues remains, thus, a major challenge for future studies. This is also complicated by the fact that no specific inhibitors for β homomers are currently available. We would, however, like to emphasize that β pentamers have received considerable attention as putative histamine gated anion channels^{4,5}.

Reviewers comment

(3) in-depth of discussion about the significance of this acidic pH-mediated regulation mechanism of β subunits in GABA transmission

Response

According to the reviewer's suggestion we discuss the possible role of pH mediated regulation mechanism of β -subunits in more detail.

See p. 11, l. 270-307 for a more extensive discussion on the possible expression of receptors and their relevance:

" β -homomers form readily in recombinant expression systems, their existence in mammalian cells is currently unknown. First evidence for picrotoxin-sensitive $I_{H(\beta x)}$ was obtained in Jurkat T cells (Supplementary Figure 5a). These cells express β -subunits at much higher abundance than α GABA_A subunits²⁸ which would be expected to facilitate formation of β homomers. It is tempting to speculate that hyperpolarizing chloride currents upon acidification counterbalances depolarization via ASIC receptors and thus may have a role in immunomodulation²⁹. However, this warrants further studies. Different activation states of T-cells might lead to variable expression of proton/ histamine sensitive and GABA sensitive receptors, respectively^{29,31,32}.

In studies on GABA-sensitive iPS cell-derived GABA neurons expressing heteropentamers we did not find evidence for $I_{H(\beta x)}$ after 4-6 days in differentiating media (Supplementary Figure 5b). These data warrant further research as the presence of heteromeric GABA_A receptors in these cells (confirmed by I_{GABA_A} , Supplementary Figure 5b) may reduce formation of homooligomeric receptors^{31,32}. It seems plausible that high expression levels of β -subunits are supportive for formation of β -homomers not only in heterologous expression systems. Given the large diversity of neuronal cell types, broad screens would be needed to potentially detect pH sensitive neuron types.

In fact, evidence for the existence of GABA insensitive receptors formed by β -subunits in neuronal preparations was obtained from studies with alternative agonists such as histamine^{5,18,44}.

Our results imply that GABA_A β receptors are activated at physiological and pathological pH changes. It is therefore tempting to speculate that such pH sensitive channels formed by ubiquitously expressed subunits in the brain may exist and serve crucial functions under acidotic conditions such as stroke and ischemia where they can enhance neuroprotective hyperpolarization. Intriguingly, in an experimental model of stroke, upregulation of GABRB3 was connected with a putative neuroprotective role⁴⁵. Although macroscopic changes in extracellular pH in the brain are tightly controlled, pH fluctuations in specific micro-domains such as the synapse might be relevant³⁰. For instance, synaptic vesicles have high proton concentrations (pH 5.7) so vesicle release during high neuronal activity could lower the pH in the synaptic cleft⁴⁶.

High levels of RNA of GABA_A beta-subunits are also found in peripheral organs and tissues with a wide range of pH or strongly fluctuating bicarbonate levels such as in endocrine tissues including pancreatic islets^{2,47}, gastrointestinal tract⁴⁸, kidneys⁴⁹, respiratory organs⁵⁰ as well as in several types of immune cells^{29,51} where a local acidification in an inflammatory locus may have relevance for their activation (Supplementary Figure 5a)⁵².

The pH sensitivity of the receptors can serve as a novel tool to analyze the tissue specific expression of homomeric β -receptors in excitable and not excitable cells. To clarify the existence of proton-gated β homomers and their physiological and pathophysiological role in neuronal and/or non-neuronal cells is thus an important goal. In tissues expressing ASIC receptors they might counterbalance a pH dependent depolarisation under acidic conditions (Supplementary Figure 5). Such receptors (which are already known to be sensitive e.g. to histamine and barbiturates) may form a class of novel potential drug targets.”

Reviewer #3 (Remarks to the Author):

This manuscript presents a new claim that GABAA receptors form proton gated chloride channels.

The authors also perform an MD simulation capturing a crucial H267-E270 interaction, many details of the said interaction are revealed by the simulation were anticipated by extensive prior mutagenesis studies coupled with functional and biochemical analysis, the direct visualization of these mechanisms is a major advance, and refreshingly the authors do a reasonable job of discussing the MD simulation in context of prior work, although this could be improved.

Also, appropriately acknowledged is the fact that it is unclear whether there could be GABA present in trace quantities during the experiment that could impact the major proton gated claim of this manuscript.

Also the authors should address the physiological relevance of the pH range tested, as far as my understanding goes the pH even in synapses seldom goes below 6.5.

Lastly I am not sure why the authors chose to ignore an array of publications related to

a) GABA-A receptor structure for e.g. PMID:30044221, PMID: 32879488 among others

b) GLICS: e.g. PMID: 30541892

A few minor points other than what's already mentioned:

Introduction, para 5, line 2: is Ref 21 the correct citation for the information provided?

Introduction Last para "Mutation H267": please correct the grammar.

Results, Section : Functional properties of proton activated chloride channels, para 2, "Even a small decrease from pH 7.2 to 6.5"- According to me between these two pH values the concentration of H₃O⁺ ion changes exponentially as pH is in log scale.

Figure 3c: readers might also benefit from having at least one Gamma subunit in the alignment, I further recommend testing the effects on a tri-heteromeric receptor and comparing the Gamma versus the Rho subunit.

Point-by-point response to reviewer #3

Reviewers comment

(1) “refreshingly the authors do a reasonable job of discussing the MD simulation in context of prior work, although this could be improved”

Response

According to the reviewers comment we have extended the discussion on MD simulations in the context of what is already known from prior work on the GLIC receptor and added a new paragraph on p.10, l. 249-264

“We propose that interaction between H267 and E270 might play a crucial role in the direct activation of homomeric GABA_A β 3 receptors by protons. Mutational studies on the GLIC channel suggests that proton activation occurs also in GLIC allosterically to the orthosteric site, but at the level of multiple loci⁴¹ and we thus cannot exclude that other residues than H267 contribute to proton-induced activation of β 3 receptors. Furthermore, additional evidence suggests that other amino acid residues may be involved in proton-induced gating of GLIC where more detailed mechanisms of receptor activation are known^{42,43}. There may be a similar picture to that of the GLIC, which has been described to have two electrostatic triads of amino acid residues in the extracellular domain (structurally highly conserved between pLGIC), which are part of a continuous network governing the gating transitions of the receptor. However, due to the absence of open/closed β 3 homomeric receptor structures, it is difficult to speculate about the exact determinants and this question remains open for further investigation. The HxxE motif we describe here is unique to the β subunits (Fig. 3c) and absent in GLIC, and is necessary and sufficient for proton sensitivity as demonstrated by the conversion mutants, irrespective of the details of transduction. While MD simulations of GLIC have been conducted⁴², a side-by-side comparison with the same method would be needed to get more insight into similarities and differences in the proton interaction sites and transduction mechanisms.”

Reviewers comment

(2) “Also, appropriately acknowledged is the fact that it is unclear whether there could be GABA present in trace quantities during the experiment that could impact the major proton gated claim of this manuscript”

Response

No traces of GABA were present in the experiments shown in Fig. 1, Fig. 2, Fig.3 and Supplementary Figure 2, Supplementary Figure 3 as there was no GABA added to the test solutions. We can completely exclude the presence of any traces of GABA.

Reviewers comment

(3) “Also the authors should address the physiological relevance of the pH range tested, as far as my understanding goes the pH even in synapses seldom goes below 6.5”

Response

This question was addressed in Fig. 2 where we illustrate $I_{H(\beta)}$ evoked by pH changes from 7.2 to 6.5.

Furthermore, physiological and pathophysiological pH changes are now discussed in more detail on p. 11, l. 270-307.

“ β -homomers form readily in recombinant expression systems, their existence in mammalian cells is currently unknown. First evidence for picrotoxin-sensitive $I_{H(\beta x)}$ was obtained in Jurkat T cells (Supplementary Figure 5a). These cells express β -subunits at much higher abundance than α GABA_A subunits²⁸ which would be expected to facilitate formation of β homomers. It is tempting to speculate that hyperpolarizing chloride currents upon acidification counterbalances depolarization via ASIC receptors and thus may have a role in immunomodulation²⁹. However, this warrants further studies. Different activation states of T-cells might lead to variable expression of proton/ histamine sensitive and GABA sensitive receptors, respectively^{29,31,32}.”

In studies on GABA-sensitive iPS cell-derived GABA neurons expressing heteropentamers we did not find evidence for $I_{H(\beta x)}$ after 4-6 days in differentiating media (Supplementary Figure 5b). These data warrant further research as the presence of heteromeric GABA_A receptors in these cells (confirmed by I_{GABA} , Supplementary Figure 5b) may reduce formation of homooligomeric receptors^{31,32}. It seems plausible that high expression levels of β -subunits are supportive for formation of β -homomers not only in heterologous expression systems. Given the large diversity of neuronal cell types, broad screens would be needed to potentially detect pH sensitive neuron types.

In fact, evidence for the existence of GABA insensitive receptors formed by β -subunits in neuronal preparations was obtained from studies with alternative agonists such as histamine^{5,18,44}.

Our results imply that GABA_A β receptors are activated at physiological and pathological pH changes. It is therefore tempting to speculate that such pH sensitive channels formed by ubiquitously expressed subunits in the brain may exist and serve crucial functions under acidotic conditions such as stroke and ischemia where they can enhance neuroprotective hyperpolarization. Intriguingly, in an experimental model of stroke, upregulation of GABRB3 was connected with a putative neuroprotective role⁴⁵. Although macroscopic changes in extracellular pH in the brain are tightly controlled, pH fluctuations in specific micro-domains such as the synapse might be relevant³⁰. For instance, synaptic vesicles have high proton concentrations (pH 5.7) so vesicle release during high neuronal activity could lower the pH in the synaptic cleft⁴⁶.

High levels of RNA of GABA_A beta-subunits are also found in peripheral organs and tissues with a wide range of pH or strongly fluctuating bicarbonate levels such as in endocrine tissues including pancreatic islets^{2,47}, gastrointestinal tract⁴⁸, kidneys⁴⁹, respiratory organs⁵⁰ as well as in several types of immune cells^{29,51} where a local acidification in an inflammatory locus may have relevance for their activation (Supplementary Figure 5a)⁵².

The pH sensitivity of the receptors can serve as a novel tool to analyze the tissue specific expression of homomeric β-receptors in excitable and not excitable cells. To clarify the existence of proton-gated β homomers and their physiological and pathophysiological role in neuronal and/or non-neuronal cells is thus an important goal. In tissues expressing ASIC receptors they might counterbalance a pH dependent depolarisation under acidic conditions (Supplementary Figure 5). Such receptors (which are already known to be sensitive e.g. to histamine and barbiturates) may form a class of novel potential drug targets.”

Reviewers comment

(4) Lastly I am not sure why the authors chose to ignore *an array of publications related to*

a) GABA-A receptor structure for e.g. PMID:30044221, PMID: 32879488 among others

b) GLICS: e.g. PMID: 30541892

Response

All references mentioned by the reviewer have been cited and discussed in the revised version of the Manuscript and Supplementary Information (Supplementary Figure 6, also see Supplementary Figure 6 in the end of the response file).

The references related to beta subunit structures are included in Supplementary Information, more specifically to align beta subunits, the structure of $\beta 1$ subunit (6DW0) from the *PMID:30044221*, of $\beta 2$ subunits (6X3T) from *PMID: 32879488* was used alongside with two $\beta 3$ subunits (4COF and 7A5V).

Their discussion (and additional discussion on GLIC touching on the *PMID: 30541892* paper) are now included into the new paragraphs on p. 10, l. 243-264.

“In order to compare the similarity of the three β subunits in the H267 - E270 region, we also examined structures with $\beta 1$ and $\beta 2$ subunits. Of these, only heteromeric structures are available^{39,40}. Thus, the individual β subunits were superposed for the best fit of the M2 segment. The amino acids of interest are structurally equivalent, and only slightly differ in sidechain rotamers. Thus, we expect the residues H267 and E270 play a structurally equivalent role in $\beta 1$ and $\beta 2$ homopentamers (Supplementary Figure 6).

We propose that interaction between H267 and E270 might play a crucial role in the direct activation of homomeric GABA_A $\beta 3$ receptors by protons. Mutational studies on the GLIC channel suggests that proton activation occurs also in GLIC allosterically to the orthosteric site, but at the level of multiple loci⁴¹ and we thus cannot exclude that other residues than H267 contribute to proton-induced activation of $\beta 3$ receptors. Furthermore, additional evidence suggests that other amino acid residues may be involved in proton-induced gating of GLIC where more detailed mechanisms of receptor activation are known^{42,43}. There may be a similar picture to that of the GLIC, which has been described to have two electrostatic triads of amino acid residues in the extracellular domain (structurally highly conserved between pGLIC), which are part of a continuous network governing the gating transitions of the receptor. However, due to the absence of open/closed $\beta 3$ homomeric receptor structures, it is difficult to speculate about the exact determinants and this question remains open for further investigation. The HxxE motif we describe here is unique to the β subunits (Fig. 3c) and absent in GLIC, and is necessary and sufficient for proton sensitivity as demonstrated by the conversion mutants, irrespective of the details of transduction. While MD simulations of GLIC have been conducted⁴², a side-by-side comparison with the same method would be needed to get more insight into similarities and differences in the proton interaction sites and transduction mechanisms.”

Reviewers comment

(5) Introduction, para 5, line 2: is Ref 21 *the correct citation for the information provided?*

Response

We have checked that the reference is correct. In addition, we added another source (now references 24 and 25).

Reviewers comment

(6) Introduction Last para “Mutation H267”: please *correct the grammar.*

Response

The grammar has been corrected, see p.4, l 58

“The mutation H267A completely abolished proton sensitivity”

Reviewers comment

(7) Results, Section : Functional properties of proton activated chloride channels, para 2, “Even a small decrease from pH 7.2 to 6.5”- According to me between these two pH values the concentration of H₃O⁺ ion changes exponentially as pH is in log scale

Response

Indeed, the word “small” was misleading as change of pH by 0.7 (from 7.2 to 6.5) corresponds to a 5-fold increase of the H⁺ concentration { $10^{0.7} \approx 5$ }. Correspondingly we removed the word small and write on p. 5, l. 105: “A decrease from pH 7.2 to 6.5 ...”.

Reviewers comment

(8) Figure 3c: readers might also benefit from *having at least one Gamma subunit in the alignment*

Response

According to the reviewer’s suggestion we have included a γ subunit into Fig. 3c. See the completed Figure 3c on p. 27, l. 637 (also see the Fig. 3 at the end of this document).

Reviewers comment

(9) I further recommend *testing the effects on a tri-heteromeric receptor* and comparing the Gamma versus the Rho subunit.

Response

According to the reviewer's suggestion we have performed these additional experiments with tri-heteromeric receptor ($\alpha 1\text{-}\beta 3\text{-}\alpha 1/\gamma 2\text{-}\beta 3$) and included the data into the Manuscript (p.5, l.113).

“Subunit concatenation ($\alpha 1\text{-}\beta 2/\beta 2\text{-}\alpha 1\text{-}\beta 2$, $\alpha 1\text{-}\beta 3/\beta 3\text{-}\alpha 1\text{-}\beta 3$ and $\alpha 1\text{-}\beta 3\text{-}\alpha 1/\gamma 2\text{-}\beta 3$) prevented both the formation of β homomers and at the same time activation of $I_{H(\beta)}$ (Supplementary Figure 4).”

See additional Supplementary Figure 4c in SI on p. 5, l. 46 (also see the Supplementary Figure 4 at the end of this document).

Reviewer #4 (Remarks to the Author):

This is an interesting report showing a novel function of GABA receptor beta subunit, although their existence in vivo has not been shown. The research is well-designed and the results are clear. I think it will be of interest to the people in the field. I want to point out several minor points to consider.

1. A potential significance of the co-expression of beta homomers and heteropentamers in the physiological and pathological conditions should be discussed, although the existence of beta homomers in mammalian cells has not yet been investigated.

2. Figure 2ab: The pH on the figure legend is 5.5. Please make an adequate correction in the manuscript. In addition, the pH used to examine I_{GABA} should be clarified in the manuscript since pH modulates GABA-activated responses on the GABAA receptors.

3. Supplementary figure 1: This figure should be the supplementary figure 2 when arranged in order of appearance. The species of construct should be presented in the figure legend.

4. Supplementary figure 2: which species of beta3 construct was used should be written in the figure legend. This figure should be the supplementary figure 1 when arranged in order of appearance.

Point-by-point response to reviewer #4***Reviewers comment***

(1) A potential significance of the co-expression of beta homomers and heteropentamers in the physiological and pathological conditions should be discussed, although the existence of beta homomers in mammalian cells has not yet been investigated.

Response

In order to address the reviewer's suggestion, we have performed additional experiments on Jurkat cells expressing predominantly GABA_A receptor β -subunits and iPS cell-derived GABANeurons expressing heteropentamers. Jurkat cells grow as a suspension and GABANeurons can be transferred to a suspension after detaching from the substrate. Both cell types were therefore suitable for studies using the planar patch clamp technique.

These new data are summarized in an additional Supplementary Figure 5 on p. 6, l. 51 (also see the Supplementary Figure 5 at the end of this document).

.

See p. 11, l. 270-307 for a more thorough discussion on the possible expression of receptors and their relevance.

" β -homomers form readily in recombinant expression systems, their existence in mammalian cells is currently unknown. First evidence for picrotoxin-sensitive $I_{H(\beta x)}$ was obtained in Jurkat T cells (Supplementary Figure 5a). These cells express β -subunits at much higher abundance than α $GABA_A$ subunits²⁸ which would be expected to facilitate formation of β homomers. It is tempting to speculate that hyperpolarizing chloride currents upon acidification counterbalances depolarization via ASIC receptors and thus may have a role in immunomodulation²⁹. However, this warrants further studies. Different activation states of T-cells might lead to variable expression of proton/ histamine sensitive and GABA sensitive receptors, respectively^{29,31,32}.

In studies on GABA-sensitive iPS cell-derived GABA neurons expressing heteropentamers we did not find evidence for $I_{H(\beta x)}$ after 4-6 days in differentiating media (Supplementary Figure 5b). These data warrant further research as the presence of heteromeric $GABA_A$ receptors in these cells (confirmed by I_{GABA} , Supplementary Figure 5b) may reduce formation of homooligomeric receptors^{31,32}. It seems plausible that high expression levels of β -subunits are supportive for formation of β -homomers not only in heterologous expression systems. Given the large diversity of neuronal cell types, broad screens would be needed to potentially detect pH sensitive neuron types.

In fact, evidence for the existence of GABA insensitive receptors formed by β -subunits in neuronal preparations was obtained from studies with alternative agonists such as histamine^{5,18,44}.

Our results imply that $GABA_A$ β receptors are activated at physiological and pathological pH changes. It is therefore tempting to speculate that such pH sensitive channels formed by ubiquitously expressed subunits in the brain may exist and serve crucial functions under acidotic conditions such as stroke and ischemia where they can enhance neuroprotective hyperpolarization. Intriguingly, in an experimental model of stroke, upregulation of GABRB3 was connected with a putative neuroprotective role⁴⁵. Although macroscopic changes in extracellular pH in the brain are tightly controlled, pH fluctuations in specific micro-domains such as the synapse might be relevant³⁰. For instance, synaptic vesicles have high proton concentrations (pH 5.7) so vesicle release during high neuronal activity could lower the pH in the synaptic cleft⁴⁶.

High levels of RNA of GABA_A beta-subunits are also found in peripheral organs and tissues with a wide range of pH or strongly fluctuating bicarbonate levels such as in endocrine tissues including pancreatic islets^{2,47}, gastrointestinal tract⁴⁸, kidneys⁴⁹, respiratory organs⁵⁰ as well as in several types of immune cells^{29,51} where a local acidification in an inflammatory locus may have relevance for their activation (Supplementary Figure 5a)⁵².

The pH sensitivity of the receptors can serve as a novel tool to analyze the tissue specific expression of homomeric β -receptors in excitable and not excitable cells. To clarify the existence of proton-gated β homomers and their physiological and pathophysiological role in neuronal and/or non-neuronal cells is thus an important goal. In tissues expressing ASIC receptors they might counterbalance a pH dependent depolarisation under acidic conditions (Supplementary Figure 5). Such receptors (which are already known to be sensitive e.g. to histamine and barbiturates) may form a class of novel potential drug targets.”

Reviewers comment

(2) Figure 2ab: The pH on the figure legend is 5.5. Please make an adequate correction in the manuscript. In addition, the pH used to examine I_{GABA} should be clarified in the manuscript since pH modulates GABA-activated responses on the GABA_A receptors.

Response

We have made an appropriate correction of the figure legend, included the pH in the subscript, see Fig. 2 on p. 25, l. 617, l.619 and l. 622.

All recordings of I_{GABA} were made at physiological pH (pH= 7.2). This is now clarified in the subscript of Fig.2 p.25, l.620, Fig.4 p.29, l.651 and Supplementary Figure 4 p.5, l.50

We also specify the pH values of the solution used to induce I_{GABA} on p.5, l.97

Reviewers comment

(3) Supplementary figure 1: This figure should be the supplementary figure 2 when arranged in order of appearance. The species of construct should be presented in the figure legend.

Response

We changed the order of the figures according to appearance in the text and specified the species in the legend, see SI p.3

Reviewers comment

(4) Supplementary figure 2: which species of beta3 construct was used should be written in the figure legend. This figure should be the supplementary figure 1 when arranged in order of appearance.

Response

According to the reviewer's suggestion we specified the species of beta3 construct in the subscript and changed the order (swap Supplementary Figure 2 with Supplementary Figure 1), see SI p.2 and p.3

1. Dionisio, L., Arias, V., Bouzat, C. & Esandi, M. del C. GABAA receptor plasticity in Jurkat T cells. *Biochimie* **95**, 2376–2384 (2013).
2. Sieghart, W. & Sperk, G. Subunit composition, distribution and function of GABA(A) receptor subtypes. *Curr Top Med Chem* **2**, 795–816 (2002).
3. Angelotti, T. P. & Macdonald, R. L. Assembly of GABAA receptor subunits: alpha 1 beta 1 and alpha 1 beta 1 gamma 2S subunits produce unique ion channels with dissimilar single- channel properties. *J. Neurosci.* **13**, 1429–1440 (1993).
4. Fleck, M. W., Thomson, J. L. & Hough, L. B. Histamine-gated ion channels in mammals? *Biochem Pharmacol* **83**, 1127–1135 (2012).
5. Nakane, T. *et al.* Single-particle cryo-EM at atomic resolution. *Nature* **587**, 152–156 (2020).

Figures (that have been added or improved in the manuscript)

Figure 2: the correct pH values in the subscript have been indicated; the pH value for I_{GABA} was specified

Fig. 2: Functional properties of proton activated homomeric β receptors.

a Activation kinetics of $I_{H(\beta_3)}$ (pH 5) through human homomeric β_3 receptors and I_{GABA} (GABA 100 μ M, pH 7.2) through $\alpha 1\beta_3$ GABA_A receptors at saturating agonist concentrations. The current traces illustrate slower activation of $I_{H(\beta_3)}$ ($t_{peak} = 1.91$ s) compared to I_{GABA} ($t_{peak} = 0.68$ s). **b** Desensitisation kinetics of $I_{H(\beta_3)}$ and I_{GABA} during long lasting exposure to pH 5 or to GABA 100 μ M. Desensitization half time for $I_{H(\beta_3)}$ is

$t_{0.5} = 58.5$ s compared to $t_{0.5} = 33.6$ s of I_{GABA} . pH in **(a, b)** was changed from 9 to 5. **c** Representative $I_{H(\beta_3)}$ induced by switching the pH from 7.2 to 6.5. **d** The pH-dependence of steady-state desensitization and of activation of receptors composed of the indicated β -subunits. Oocytes were conditioned for 10 minutes at the indicated pH, and receptors were subsequently activated by shifting the pH rapidly to 4.5 in order to estimate the the fraction of available (not yet desensitized) channels. The normalized current response is plotted as a function of the conditioning pH (superimposed data for current activation (dash lines) are taken from Fig. 1). Half-maximal desensitisation occurs at pH 6.4 ± 0.02 for $\beta_1(S265N)$, at pH 6.97 ± 0.06 for β_2 and at pH 6.6 ± 0.15 ($n = 4$) for β_3 homomers. pH was reduced from 9 to the indicated values. The data are presented as mean values \pm SEM, $n=5$.

Figure 3: $\gamma 2$ subunit was included in the alignment (Fig.3c)

Fig. 3: Loss of function ($\beta 3(\text{H267A})$) mutation in GABA_A subunit homomers and a hydrogen bond formation between H267 and E270.

a Location of H267 in the pore region of the crystal structure of $\beta 3$ homomers (the position of H267 is indicated with violet spheres)⁹. **b** Mutation H267A in the human subunits $\beta 3$ subunit prevented activation of $I_{H(\beta 3(\text{H267A}))}$ but, instead, induced pH-dependent inhibition of large baseline currents (upper traces).

Dotted line illustrates zero current level. Lower traces illustrate retained sensitivity for other agonists (100 μ M pentobarbital, 1mM histamine and 100 μ M bicuculline). **c** Sequence alignment of the transmembrane M2 α -helix of α 1, β 1-3 (human), γ 2 (human) and ρ 1 (rat) subunits (segment does not differ between rat and human). The location of H267 and E270 in β subunits and the corresponding homologous positions in the ρ 1 subunit (double mutant ρ 1(G331H-A334E)) are illustrated. **d** Histogram of the number of hydrogen bonds between protonated and deprotonated H267 and E270. **e** Snapshots of the ring structure observed in the MD runs.

Supplementary Figure 4: additional experiments with the tri-heteromeric receptor ($\alpha 1\text{-}\beta 3\text{-}\alpha 1/\gamma 2\text{-}\beta 3$) were performed, the results are included in the Supplementary Figure 4c.

Supplementary Figure 4. Concatenation of subunits prevents activation of $I_{H(\beta)}$. GABA_A receptors composed of concatenated subunits $\alpha 1\text{-}\beta 2/\beta 2\text{-}\alpha 1\text{-}\beta 2$ (rat) (a), $\alpha 1\text{-}\beta 3/\beta 3\text{-}\alpha 1\text{-}\beta 3$ (rat) (b) and $\alpha 1\text{-}\beta 3\text{-}\alpha 1/\gamma 2\text{-}\beta 3$ (rat) (c) are not activated by changing the pH from 9 to indicated values (a, b and c, respectively) but activated by GABA (pH 7.2).

Supplementary Figure 5: this figure was newly introduced

Supplementary Figure 5. Proton- and GABA-induced currents in Jurkat cells (a) and iPS cell-derived iCell GABAergic neurons (b).

a Representative currents recorded from Jurkat cells upon shifting the pH from 7.2 to 5 alone (left) or in the presence of 200 μM amiloride (middle overlaid currents, blue). Application of pH 5 produced ASIC-like fast activating currents, whereas co-application of pH 5 with amiloride (an ASIC blocker²) resulted in slower activating and desensitising currents that were completely blocked by picrotoxin (100 μM). Application of GABA did not activate I_{GABA} in these cells. Average mean current amplitudes are 440 ± 118 pA for pH 5-induced currents and 34 ± 4 pA for pH 5 in presence of 200 μM amiloride ($n = 11$, the data are presented as mean values \pm SEM).

b Representative currents recorded from iPS cell-derived iCell GABAergic neurons cells. Application of pH 5 resulted in a current (left) which was completely abolished by co-application with amiloride (middle trace), consistent with an ASIC current. I_{GABA} (induced by 100 μM GABA) is shown on the right (indicating the presence of heteromeric $GABA_A$ receptors in these neurons). Average current amplitudes are 225 ± 44 pA for pH 5-induced ASIC currents and 308 ± 35 pA for GABA-induced currents ($n = 5$, the data are presented as mean values \pm SEM).

Supplementary Figure 6: this figure was newly introduced

Supplementary Figure 6. Alignment of the M2 segment of different experimental structures of beta subunits. The protein is represented as cartoon; residues H267 and E270 (numbering corresponds to $\beta 3$ numbers) are shown as sticks. The $\beta 3$ subunits are colored in slate blue (4COF³) and dark blue (7A5V⁴), $\beta 1$ subunit in grey (6DW0⁵) and the $\beta 2$ subunits of the cryo-EM structure 6X3T⁶ in light orange (chain A) and dark orange (chain E). Positions of H267 and E270 are equivalent and only slight differences in the rotameric state of are visible. We therefore expect these residues play a structurally equivalent role in homopentamers of $\beta 1$ and $\beta 2$. In the heteropentamers, the TMD adopts a different conformation, which limits this structural comparison due to the lacking structure of homomeric $\beta 1$ or $\beta 2$ structures.

REVIEWERS' COMMENTS:

Reviewer #2 (Remarks to the Author):

Previous concerns have been addressed.

Reviewer #3 (Remarks to the Author):

I agree with the authors addition of new body of data, and it addresses any concerns that I had. The manuscript is in a much better shape and provides a more comprehensive view of the question at hand. Please accept my congratulations on this wonderful piece of work.

Reviewer #4 (Remarks to the Author):

The authors have addressed all of my comments. This manuscript is now suitable for publication.